# EGG-SR: Embedding Symbolic Equivalence into Symbolic Regression via Equality Graph

**Nan Jiang**
University of Texas - El Paso
njiang@utep.edu

**Ziyi Wang, Yexiang Xue**
Purdue University
{wang4538,yexiang}@purdue.edu

## Abstract

Symbolic regression seeks to uncover physical laws from experimental data by searching for closed-form expressions, which is an important task in AI-driven scientific discovery. Yet the exponential growth of the search space of expression renders the task computationally challenging. A promising yet underexplored direction for reducing the search space and accelerating training lies in *symbolic equivalence*: many expressions, although syntactically different, define the same function – for example, $\log(x_1^2 x_2^3)$, $\log(x_1^2) + \log(x_2^3)$, and $2\log(x_1) + 3\log(x_2)$. Existing algorithms treat such variants as distinct outputs, leading to redundant exploration and slow learning. We introduce EGG-SR, a unified framework that integrates symbolic equivalence into a class of modern symbolic regression methods, including Monte Carlo Tree Search (MCTS), Deep Reinforcement Learning (DRL), and Large Language Models (LLMs). EGG-SR compactly represents equivalent expressions through the proposed EGG module (via equality graphs), accelerating learning by: (1) pruning redundant subtree exploration in EGG-MCTS, (2) aggregating rewards across equivalent generated sequences in EGG-DRL, and (3) enriching feedback prompts in EGG-LLM. Theoretically, we show the benefit of embedding EGG into learning: it tightens the regret bound of MCTS and reduces the variance of the DRL gradient estimator. Empirically, EGG-SR consistently enhances a class of symbolic regression models across several benchmarks, discovering more accurate expressions within the same time limit.

Project page is at: `https://nan-jiang-group.github.io/egg-sr`.

## 1 Introduction

Symbolic regression aims to automatically discover physical knowledge from experimental data and has been widely used in scientific domains (Schmidt & Lipson, 2009; Udrescu & Tegmark, 2020; Cory-Wright et al., 2024; Yu & Wang, 2024; LaFollette et al., 2025). Many contemporary methods for symbolic regression formulate the search for optimal expressions as a sequential decision-making process. In literature, existing methods learn to predict the sequence of grammar rules (Sun et al., 2023), the traversal sequence of expression trees (Petersen et al., 2021; Kamienny et al., 2022), or executable strings that follow Python syntax (Shojaee et al., 2025; Zhang et al., 2025). This task remains computationally challenging due to its NP-hard nature (Virgolin & Pissis, 2022), that is, the search space of candidate expressions grows exponentially with the data dimension.

A promising yet underexplored direction for reducing the search space and accelerating discovery is the integration of *symbolic equivalence* into learning algorithms. For example, these expressions $\log(x_1^2 x_2^3)$, $\log(x_1^2) + \log(x_2^3)$, and $2\log(x_1) + 3\log(x_2)$ all represent the same math function and are therefore *symbolically equivalent*. Ideally, a well-trained model would recognize such equivalence and assign identical goodness-of-fit, rewards, or losses to the corresponding predicted expressions (Allamanis et al., 2017), since these expressions produce identical functional outputs and attain the same prediction error on the dataset. In the literature, existing SR algorithms treat these expressions as distinct outputs, leading to redundant exploration of the search space and slow training. The main challenge of this direction is: how to represent symbolically-equivalent expressions and embed them into modern learning frameworks in a *unified and scalable* manner?

Since the number of equivalent variants grows exponentially with expression length, explicitly maintaining the set of equivalent expressions is not time and memory scalable. To mitigate this scalability challenge, a line of recent works introduced the Equality graph (e-graph), a data structure that compactly encodes the set of equivalent variants by storing shared sub-expressions only once (Nandi et al., 2021; Willsey et al., 2021; Kurashige et al., 2024). e-graphs have been applied to tasks, including program optimization (Barbulescu et al., 2024), dataset generation of equivalent expressions (Zheng et al., 2025). In genetic programming-based symbolic regression, e-graphs have been used for duplicate detection (de França & Kronberger, 2025), expression simplification (de França & Kronberger, 2023), and expression template pattern matching (de França & Kronberger, 2025). Despite the empirical successes, we find that a unified framework that enables diverse symbolic regression algorithms to interact with e-graphs to accelerate learning has room for improvement.

We present a unified framework, EGG-SR, that integrates symbolic equivalence into a class of learning algorithms via e-graphs. Our framework encompasses EGG-MCTS, EGG-DRL, and EGG-LLM. The core idea is to leverage EGG to efficiently sample equivalent variants of expressions predicted by SR algorithms and compute a new equivalence-aware learning objective. Specifically, (1) EGG-MCTS prunes redundant exploration over equivalent subtrees; (2) EGG-DRL aggregates rewards over equivalent expressions, stabilizing training; (3) EGG-LLM enriches the feedback prompt with multiple equivalent expressions to better guide next round predictions. Under mild theoretical assumptions, we show the benefit of embedding symbolic equivalence into learning: (1) EGG-MCTS offers a tighter regret bound than standard MCTS (Sun et al., 2023), and (2) the gradient estimator of EGG-DRL exhibits a lower variance than that of standard DRL (Petersen et al., 2021).

In experiments, we evaluate EGG-SR with several representative symbolic regression baselines across several challenging benchmarks. We demonstrate its advantages over existing approaches using EGG than without. EGG consistently improves performance across diverse frameworks, discovering more accurate expressions than baseline within a fixed time budget.

## 2 PRELIMINARIES

**Symbolic Expression.** Let $\mathbf{x} = (x_1, \ldots, x_n)$ denote input variables and $\mathbf{c} = (c_1, \ldots, c_m)$ be coefficients. A symbolic expression $\phi$ connects these variables and coefficients using mathematical operators such as addition, multiplication, and logarithm. For example, $\phi = 3\log x_1 + 2\log x_2$ is a symbolic expression composed of variables $\{x_1, x_2\}$, operators $\{+, \log\}$, and coefficients $\{c_1 = 3, c_2 = 2\}$. In literature, symbolic expressions have been represented as binary trees (de França & Kronberger, 2025), pre-order traversal sequences of the binary tree (Petersen et al., 2021), topological traversal sequences of the expression graph (Kahlmeyer et al., 2024; Xiang et al., 2025), or sequences of production rules defined by a context-free grammar (Sun et al., 2023).

To handle all symbolic objects in this work, a context-free grammar is adopted to represent symbolic expressions (Brence et al., 2021). The grammar is defined by a tuple $\langle V, \Sigma, R, S \rangle$ where (1) $V$ is a set of *non-terminal* symbols representing arbitrary sub-expressions, i.e., $V = \{A\}$; (2) $\Sigma$ is a set of *terminal* symbols, including input variables and coefficients, i.e., $\{x_1, \ldots, x_n\} \cup \{\mathtt{c}\}$; (3) $R$ is a set of production rules representing mathematical operations. For example, $A \to A \times A$ denotes multiplication, and the semantics is to replace the left-hand side with the right-hand side; (4) $S$ is the start symbol, typically set to $S = A$. Given a sequence of production rules, an expression is constructed by sequentially applying each rule to the *leftmost* nonterminal, starting from $S$. If the resulting string contains no nonterminals, it corresponds to a valid expression.

Figure 6 (in appendix) shows the expression construction with sequence $(A \to A \times A, A \to \mathtt{c}, A \to \log(x_1))$. The first rule $A \to A \times A$ expands the start symbol $\phi = A$ to $\phi = A \times A$. Applying the next rule $A \to \mathtt{c}$ yields $\phi = c_1 \times A$. An index is assigned to the coefficient symbol $\mathtt{c}$, to differentiate multiple coefficients. Finally, applying $A \to \log(x_1)$ to $\phi = c_1 \times A$ yields $\phi = c_1 \log(x_1)$.

**Symbolic Equivalence under a Rewrite System.** Rewrite rules are widely used to simplify, rearrange, and reformulate expressions in tasks such as code optimization and automated theorem proving (Huet & Oppen, 1980; Nandi et al., 2021). A rewrite system provides a principled procedure for transforming expressions by replacing sub-expressions according to predefined patterns.

Formally, a rewrite rule $r_i$ is written as "LHS $\leadsto$ RHS", where the left-hand side (LHS) specifies a pattern to be *matched*, and the right-hand side (RHS) specifies the *substitution* applied upon a match.

Given a set of rewrite rules $\mathcal{R} = \{r_1, r_2, r_3, \ldots\}$, the *symbolic equivalence* relation $\equiv_{\mathcal{R}}$ induced by $\mathcal{R}$ is defined as follows: for two symbolic expressions $\phi_1$ and $\phi_2$,

$$\phi_1 \equiv_{\mathcal{R}} \phi_2 \qquad \text{if and only if} \qquad \phi_1 \Rightarrow^* \phi_2 \ \text{ or } \ \phi_2 \Rightarrow^* \phi_1, \qquad (1)$$

where $\phi_1 \Rightarrow^* \phi_2$ means that $\phi_1$ can be transformed into $\phi_2$ by applying a finite sequence of rewriting using $\mathcal{R}$. In other words, two expressions are equivalent under $\mathcal{R}$ if one can be transformed into the other via repeated rewriting.

For example, consider the rewrite rule $\log(a \times b) \rightsquigarrow \log(a) + \log(b)$ (denoted as $r_1$), where $a$ and $b$ are placeholders for arbitrary sub-expressions. Since $\phi_1 = \log(x_1^3 x_2^2)$ can be transformed into $\phi_2 = \log(x_1^3) + \log(x_2^2)$ by applying $r_1$ once, $\phi_1$ and $\phi_2$ are *symbolically equivalent* under $r_1$.

In this work, the known mathematical identities listed in Table 3 (in the appendix) are encoded as rewrite rules in $\mathcal{R}$. Section 3.1 applies these rewrite rules over the e-graph to generate a batch of symbolically equivalent expressions via matching (Figure 1b) and substitution (Figure 1c) operations. Implementation details of rewrite rules are provided in Appendix B.2.

**Symbolic Regression (SR)** posits that experimental data are generated by an underlying closed-form expression, an assumption that is widely adopted across the sciences (Ma et al., 2022). Given a dataset $D = \{(\mathbf{x}_i, y_i)\}_{i=1}^N$, the goal is to find an optimal expression $\phi^*$ that minimizes the loss:

$$\phi^* \leftarrow \arg\min_{\phi} \ \frac{1}{N} \sum_{i=1}^N \ell(\phi(\mathbf{x}_i, \mathbf{c}), y_i),$$

where function $\ell$ measures the discrepancy between the prediction $\phi(\mathbf{x}_i, \mathbf{c})$ and the ground truth $y_i$. The coefficients $\mathbf{c}$ in $\phi$ are typically optimized on the training data $D$ using numerical optimizers such as BFGS (Fletcher, 2000). This problem is NP-hard (Virgolin & Pissis, 2022), posing a major challenge for SR algorithms. Recent efforts to mitigate this challenge are reviewed in Section 4.

## 3 METHODOLOGY

### 3.1 EGG: EQUALITY GRAPH FOR GRAMMAR-BASED SYMBOLIC EXPRESSION

Enumerating all equivalent variants of a symbolic expression is combinatorially expensive. Storing these variants explicitly is time-consuming and memory-inefficient. To mitigate this scalability challenge, we adopt the recently proposed Equality graph (i.e, E-graph) data structure (Willsey et al., 2021; Waldmann et al., 2022), which compactly represents the set of equivalent expressions by sharing common subexpressions. We extend E-graphs to support grammar-based symbolic expressions, noted as EGG, facilitating unified integration with symbolic regression algorithms.

**Definition.** An *e-graph* consists of a collection of equivalence classes, called *e-classes*. Each e-class contains a set of *e-nodes* representing symbolically equivalent sub-expressions (Willsey et al., 2021). Each e-node encodes a mathematical operation and references a list of child e-classes corresponding to its operands. Edges always point from an e-node to e-classes.

Figure 1(d) shows an example e-graph. The color-highlighted part is an e-class (in dashed box) containing two e-nodes (in solid boxes): $A \rightarrow \log(A)$ and $A \rightarrow A + A$. The two e-nodes represent logarithmic operation and addition, respectively. The e-node $A \rightarrow \log(A)$ has a single outgoing edge to its child e-class $A \rightarrow A \times A$, because $\log(\cdot)$ operator is unary.

**Construction.** In this work, an e-graph is initialized with a sequence of production rules, representing an input expression. Each rewrite rule (as defined in section 2) is applied by *matching* its left-hand side (LHS) pattern against the current e-graph, which involves traversing all e-classes to identify subexpressions that match LHS. For every successful match, new e-classes and e-nodes corresponding to the right-hand side (RHS) of the rule are created. A *merge* operation is applied to incorporate the new e-class with the matched e-class, thereby preserving the structure of known equivalences. This process, known as *equality saturation*, iteratively applies pattern matching and merging until either no further rules can be applied or a maximum number of iterations is reached.

**Example 3.1.** Figure 1 shows an example e-graph construction with the rewrite rule $\log(a \times b) \rightsquigarrow \log(a) + \log(b)$, where $a$ and $b$ are placeholders for arbitrary sub-expressions. The e-graph is initialized with an expression $\phi = \log(x_1^3 x_2^2)$ in Figure 1(a). The LHS is *matched* against the color-highlighted e-classes in Figure 1(b) with $a = x_1^3$ and $b = x_2^2$. The *substitution* step constructs

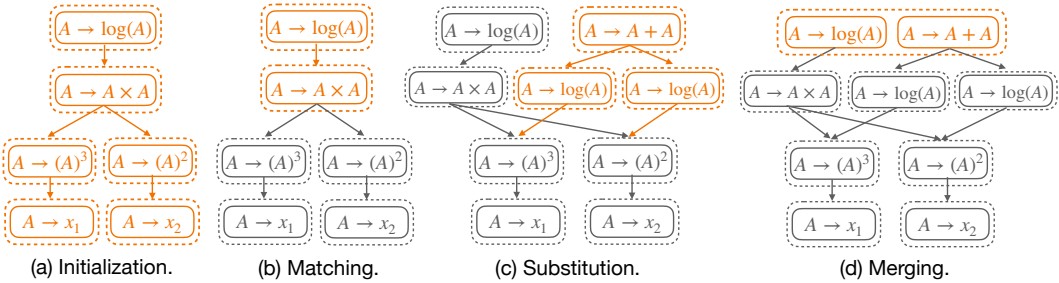

| (a) Initialization. | (b) Matching. | (c) Substitution. | (d) Merging. |

Figure 1: Example e-graph construction by applying rewrite rule $\log(a \times b) \rightsquigarrow \log(a) + \log(b)$ to an e-graph representing the expression $\log(x_1^3 x_2^2)$. **(a)** The initialized e-graph consists of *e-classes* (dashed boxes), each containing equivalent *e-nodes* (solid boxes). Edges connect e-nodes to their child e-classes. **(b)** The matching step identifies the e-nodes that match the LHS of the rule. **(c)** The substitution step adds new e-classes and edges corresponding to the RHS to the e-graph. **(d)** The merging step consolidates equivalent e-classes. See Example 3.2 for a detailed explanation.

e-classes and e-nodes that represent RHS, which are color-highlighted in Figure 1(c). The newly created e-class is *merged* with the matched e-class in Figure 1(d). The resulting e-graph represents two equivalent expressions: $\log(x_1^3 x_2^2)$ and $\log(x_1^3) + \log(x_2^2)$. The e-graph in Figure 1(d) saves memory by storing two sub-expressions $x_1^3$ and $x_2^2$ only once. Additional EGG visualizations on more complex expressions are provided in Appendix D.1.

**Extraction.** After the e-graph is saturated, an extraction step is performed to obtain $K$ representative expressions (Goharshady et al., 2024). Because an e-graph encodes up to an exponential number of equivalent expressions, exhaustive enumeration is computationally infeasible. We therefore adopt two practical strategies: (1) *cost-based extraction*, which selects several simplified expressions by minimizing a user-defined cost function over operators and variables (de França & Kronberger, 2025), and (2) *random-walk sampling*, which generates a batch of expressions by stochastically traversing the e-graph. A detailed explanation of extraction is provided in Appendix B.3.2.

**Interaction with SR algorithms.** Prior research has leveraged e-graphs, based on the principle of Occam's razor, to obtain the most simplified and least-cost equivalent form (de Franca & Kronberger, 2025). In this study, however, equivalence-aware learning (detailed in Section 3.2) is encouraged by explicitly exposing SR algorithms to an extra subset of equivalent variants.

The EGG module is primarily used to generate a subset of equivalent variants—i.e., expressions that represent the same mathematical function under a set of math identities. Given a sequence of production rules predicted by an SR algorithm, EGG constructs the e-graph and then performs extraction (via random-walk sampling) to return a batch of equivalent sequences. EGG is implemented for *grammar-based* symbolic expressions in pure Python, following the original e-graph paper (Willsey et al., 2021). Implementation of EGG is provided in Appendix B.

## 3.2 EMBEDDING SYMBOLIC EQUIVALENCE INTO SYMBOLIC REGRESSION VIA EGG

**Embedding EGG into Monte Carlo Tree Search.** MCTS (Sun et al., 2023; Ruan et al., 2025) maintains a search tree to explore an optimal sequence of decisions, here corresponding to a sequence of production rules. Each edge is labeled with a production rule, and each node is labeled by the sequence of edge labels from the root. By the grammar definition, this node label corresponds to a partially completed (or complete) expression.

During learning, MCTS iterates the following four steps (Brence et al., 2021): (1) *Selection*. Starting from the root node, successively select the edge of node $s$ (noted as $a$) with the highest Upper Confidence Bound for Trees (UCT) (Kocsis & Szepesvári, 2006):

$$\text{UCT}(s, a) = \text{reward}(s, a) + \alpha \sqrt{\log(\text{visits}(s))/\text{visits}(s, a)} \qquad (2)$$

where $\text{reward}(s, a)$ is the average reward obtained by selecting edge $a$ at node $s$, $\text{visits}(s)$ is the number of visits to node $s$, and $\text{visits}(s, a)$ is the number of times rule $a$ has been selected at node $s$. The constant $\alpha$ (often set to $\sqrt{2}$ in theory) balances between exploration and exploitation. (2)

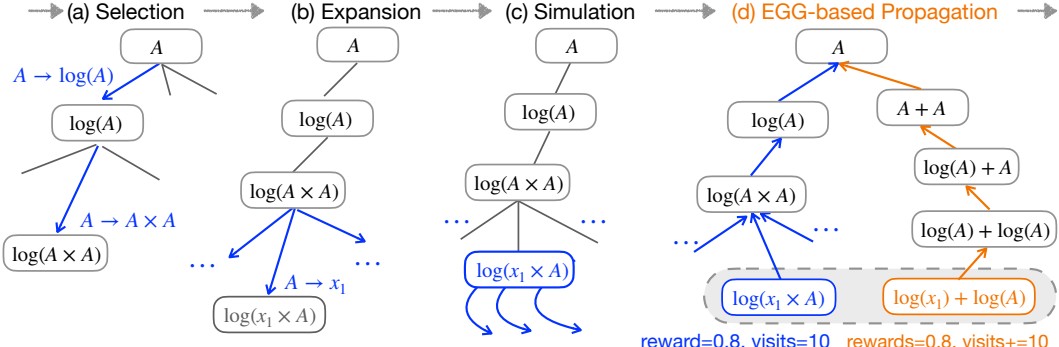

Figure 2: The EGG-MCTS pipeline consists of: **(a)** Starting at the root node, MCTS selects the child with the highest UCT score (in equation 2) until reaching a leaf. **(b)** The selected leaf produces new child nodes by applying all applicable production rules. **(c)** For each child, run several rollouts to complete the expression template by sampling additional rules. The resulting expressions are fitted to the data to estimate their coefficients. **(d)** Rewards and visit counts from the selected leaf are back-propagated to the root. In addition to updating the selected path (**blue**), we also update those equivalent paths (**orange**) identified by our EGG module.

*Expansion*. When reaching a leaf node $s$, expand the search tree by generating its child nodes using all production rules. (3) *Simulation*. Perform several rollouts for each child to evaluate the average reward of node $s$. In each rollout, generate a valid expression $\phi$ by randomly applying production rules until completion. Then, estimate the optimal coefficients in $\phi$ and evaluate its reward. A common reward function is $1/(1 + \text{NMSE}(\phi))$. (4) *Backpropagation*. Update the reward estimates and visit counts for node $s$ and all its parents up to the root node. After the final iteration, MCTS returns the expression with the highest reward encountered during training as its prediction.

EGG-*based Backpropagation.* Our backpropagation strategy is motivated by transposition tables (Childs et al., 2008; Leurent & Maillard, 2020), which use a table to cache identical nodes (e.g., via hashing) in a search tree and share their statistics during training. This mechanism propagates information as if the search had visited all identical nodes. Such tables are effective in domains such as Go and Hearthstone, where nodes that are identical can be easily determined. In symbolic regression, however, two nodes may be identical only up to symbolic equivalence induced by rewrite rules, so a hashing-based transposition table is not directly applicable. To address this, EGG is used to identify equivalent paths and nodes.

Concretely, the path—representing a partially completed expression—is first converted into an initial e-graph. The e-graph is then saturated by repeatedly applying the set of rewrite rules. From this saturated e-graph, we sample several distinct equivalent sequences and check if the tree contains corresponding paths. If so, we apply backpropagation to all of them. In this way, we avoid redundant exploration by sharing the rewards and visit counts of equivalent paths and nodes.

This modification in EGG-MCTS changes the interpretation of equation (2). $\text{visits}(s)$ no longer counts how many times the specific tree node $s$ appears on simulated paths; instead, it counts visits to any representative within the associated equivalence class. Conceptually, this mirrors the transposition table that shares statistics across identical tree nodes.

In Theorem 3.1, we show that EGG-MCTS accelerates learning by reducing the search tree's effective branching factor relative to standard MCTS. It prevents redundant exploration of equivalent subtrees and concentrates sampling on genuinely distinct (and potentially near-optimal) paths.

**Example 3.2.** Figure 2 shows an example execution pipeline of EGG-MCTS. Specifically, Figure 2(d) highlights two distinct root-to-node paths in the search tree:

Path 1 : $(A \to \log(A), A \to A \times A, A \to x_1)$,         Node $s_1$: $\log(x_1 \times A)$.

Path 2 : $(A \to A + A, A \to \log(A), A \to x_1, A \to \log(A))$,     Node $s_2$: $\log(x_1) + \log(A)$.

Here, each path is a sequence of edge labels from the root to the leaf node $s_i$. Based on the grammar definition in section 2, node $s_1$ corresponds to the partially completed expression $\log(x_1 \times A)$, while node $s_2$ represents $\log(x_1) + \log(A)$. The two nodes are equivalent under the rewrite rule

$\log(ab) \rightsquigarrow \log a + \log b$. Consequently, their rewards, estimating the averaged goodness-of-fit of expressions on the training data, should be approximately equal:

$$\texttt{reward}(s_1, a) \approx \texttt{reward}(s_2, a), \qquad \forall a \in \text{ the set of production rules}$$

Standard MCTS would explore the subtrees rooted at $s_1$ and $s_2$ independently, because it is unaware of their equivalence. This results in redundant computation and slows down learning. With EGG-MCTS, the visit counts and reward estimates of both paths are updated simultaneously, eliminating the need for extra iterations on the orange leaf in Figure 2(d). See Example B.1 for the case of other rewrite rules, e.g., $\sin^2(a) + \cos^2(a) \rightsquigarrow 1$ and $a/a \rightsquigarrow 1$.

**Embedding EGG into Deep Reinforcement Learning.** DRL typically employs a neural sequential decoder to predict an expression by sampling a sequence of production rules from the model distribution. The reward assigns higher values to expressions that better fit the training data (Petersen et al., 2021; Landajuela et al., 2022; Jiang et al., 2024). The pipeline of DRL and EGG-DRL is presented in Figure 8.

At every step, the sequential decoder samples the next production rule from a distribution over all available rules, conditioned on the previously generated sequence. The decoder thus induces a distribution $p_\theta(\tau)$ over the rule sequence $\tau$. The reward function is typically defined as $\texttt{reward}(\tau) = 1/(1 + \texttt{NMSE}(\phi))$, where $\phi$ is the expression constructed by $\tau$ following grammar definition (in section 2). The learning objective is to maximize the expected reward of generated expressions on the training data: $\mathbb{E}_{\tau \sim p_\theta} [\texttt{reward}(\tau)]$, whose gradient is $\mathbb{E}_{\tau \sim p_\theta} [\texttt{reward}(\tau) \nabla_\theta \log p_\theta(\tau)]$. In practice, the gradient is approximated via Monte Carlo. Sampling $N$ sequences $\{\tau_1, \ldots, \tau_N\}$ from the decoder, the policy gradient estimator computes:

$$g(\theta) \approx \frac{1}{N} \sum_{i=1}^{N} (\texttt{reward}(\tau_i) - b) \nabla_\theta \log p_\theta(\tau_i), \tag{3}$$

where $b$ is a baseline used to reduce variance (Weaver & Tao, 2001). Recent work (Petersen et al., 2021) further proposes using a top-quantile subset of samples, rather than the sample mean.

EGG-*based Policy Gradient Estimator.* For each sampled sequence $\tau_i$, we construct an e-graph that compactly encodes all of its equivalent expressions. From this e-graph, we sample $K - 1$ equivalent sequences $\{\tau_i^{(2)}, \ldots, \tau_i^{(K)}\}$. We then revise the policy-gradient estimator as

$$g_{\texttt{egg}}(\theta) \approx \frac{1}{N} \sum_{i=1}^{N} \left(\texttt{reward}(\tau_i) - b'\right) \nabla_\theta \log \left[ \sum_{k=1}^{K} p_\theta(\tau_i^{(k)}) \right], \tag{4}$$

where $\tau_i^{(1)}$ is the original sequence $\tau_i$ and $b'$ is the corresponding baseline, and $\sum_{k=1}^{K} p_\theta(\tau_i^{(k)})$ aggregates the probabilities of all equivalent sequences that share the same reward. In Theorem 3.2, we show that EGG improves DRL training by yielding a lower-variance gradient estimator than standard DRL (Petersen et al., 2021).

**Embed EGG into Large-Language Model.** LLM is applied to search for optimal symbolic expressions with prompt tuning (Merler et al., 2024; Shojaee et al., 2025). The procedure consists of three key steps: (1) *Hypothesis Generation*: The LLM generates multiple candidate expressions based on a prompt describing the problem background and the definitions of each variable. (2) *Data-Driven Evaluation*: Each candidate expression is evaluated based on its fitness on the training dataset. (3) *Experience Management*: In subsequent iterations, the LLM receives feedback in the form of previously predicted expressions and their corresponding fitness scores, allowing it to refine future generations. High-fitness expressions are retained and updated over multiple rounds of iteration.

EGG-*based Feedback Prompt.* Since LLMs typically generate Python functions rather than symbolic expressions directly, we introduce a wrapper that parses the generated Python code into symbolic expressions. These expressions are then transformed into e-graphs using a set of rewrite rules. From each e-graph, we extract semantically equivalent expressions and summarize them into a similar feedback message, which is incorporated into the prompt for the next round. This augmentation enables the LLM to observe a richer set of functionally equivalent expressions, potentially improving the quality and accuracy of predictions in future iterations.

## 3.3 CONNECTION TO EXISTING METHODS

Prior work has explored alternative expression representations based on layer-wise symbolic networks (SymNet) (Sahoo et al., 2018; Li et al., 2024), which are not directly compatible with our grammar-based formulation. Recent studies on SymNet further show that many learned coefficients can be aggregated or merged (Wu et al., 2024). Extending this notion of coefficient equivalence to *sub-expression equivalence* within SymNet remains an interesting open problem.

For DRL-based approaches, several extensions of the original method (Petersen et al., 2021) have been proposed, including (Mundhenk et al., 2021; Landajuela et al., 2022). An important open question is whether symbolic-equivalence can be integrated into these extensions in a compatible and effective manner, and whether doing so can further improve overall performance.

Finally, Kamienny et al. (2022); Shojaee et al. (2023) encode data directly with a Transformer and predict an expression traversal sequence end-to-end under a cross-entropy objective. A natural way to incorporate EGG is to use it during training to generate multiple equivalent, correct target sequences. How to best leverage EGG at inference time, however, remains an open problem.

## 3.4 THEORETICAL JUSTIFICATION ON EGG-SR ACCELERATING LEARNING

Theorem 3.1 shows that EGG-MCTS achieves an asymptotically tighter regret bound than standard MCTS, as the effective branching factor satisfies $\kappa_\infty \leq \kappa$. Intuitively, by identifying symbolically equivalent nodes and sharing their search statistics, EGG prevents redundant exploration of equivalent subtrees and concentrates sampling on genuinely distinct (and potentially near-optimal) paths. After many iterations, EGG-MCTS concentrates more quickly on the near-optimal region of the search space, which is captured by a smaller effective branching factor.

Also, Theorem 3.2 shows that embedding EGG into DRL produces an unbiased gradient estimator while strictly reducing gradient variance, ensuring more stable and efficient policy updates.

**Theorem 3.1.** Consider embedding EGG into the MCTS framework. Given Definitions 1 and 3 (in appendix), let $n$ denote the total number of learning iterations, $\gamma \in (0, 1)$ the discount factor of the corresponding Markov decision process, $\kappa$ be the near-optimal branching factor of standard MCTS, and $\kappa_\infty$ the corresponding branching factor of EGG-MCTS. Then the regret bounds satisfy

$$\texttt{regret}(n) \;=\; \widetilde{\mathcal{O}}\left(n^{-\frac{\log(1/\gamma)}{\log \kappa}}\right), \qquad \texttt{regret}_{\texttt{egg}}(n) \;=\; \widetilde{\mathcal{O}}\left(n^{-\frac{\log(1/\gamma)}{\log \kappa_\infty}}\right), \qquad \text{with } \kappa_\infty \leq \kappa.$$

*Proof Sketch.* Leurent & Maillard (2020) analyze MCTS on a graph obtained by merging identical tree nodes and sharing their statistics. Their analysis *unrolls* the graph into a tree that contains all graph-traversable paths. The search tree in EGG-MCTS behaves identically to the unrolled tree. Our final results follow their regret analysis on the unrolled tree. A detailed proof is in Appendix A.2. □

**Theorem 3.2.** Consider embedding EGG into the DRL framework. Let $\tau \sim p_\theta$ denote trajectories sampled from the distribution $p_\theta$, and consider the two estimators defined in Equations (3) and (4). *(1) Unbiasedness.* The expectation of the standard estimator $g(\theta)$ equals that of the EGG-based estimator $g_{\texttt{egg}}(\theta)$: $\mathbb{E}_{\tau \sim p_\theta}\big[g(\theta)\big] = \mathbb{E}_{\tau \sim p_\theta}\big[g_{\texttt{egg}}(\theta)\big]$. *(2) Variance Reduction.* The variance of the proposed estimator is smaller than that of the standard estimator:

$$\mathbb{V}\mathrm{ar}_{\tau \sim p_\theta}\big[g_{\texttt{egg}}(\theta)\big] \leq \mathbb{V}\mathrm{ar}_{\tau \sim p_\theta}\big[g(\theta)\big].$$

*Proof Sketch.* For *(1)*, unbiasedness can be obtained by expanding the definitions of $g(\theta)$ and $g_{\texttt{egg}}(\theta)$. For *(2)*, the key observation is that EGG groups together equivalent trajectories that share the same reward. Averaging over sequences with identical rewards reduces within-group variability, which yields a smaller variance. A full proof is provided in Appendix A.3. □

## 4 RELATED WORKS

**Knowledge-Guided Scientific Discovery.** Recent efforts have explored incorporating physical and domain-specific knowledge to accelerate symbolic discovery. AI-Feynman (Udrescu & Tegmark, 2020; Udrescu et al., 2020; Keren et al., 2023; Cornelio et al., 2023) constrained the search space to

expressions that exhibit compositionality, additivity, and generalized symmetry. Similarly, Tenachi et al. (2023) encoded physical unit constraints into learning to eliminate physically impossible solutions. Other works further constrained the search space by integrating user-specified hypotheses and prior knowledge, offering a guided approach to symbolic regression (Bendinelli et al., 2023; Kamienny, 2023; Shojaee et al., 2025; Taskin et al., 2026; Zhang et al., 2025). Our EGG-SR presents a new idea in knowledge-guided learning that is orthogonal to existing approaches.

**Symbolic Equivalence** is a central concept in program synthesis and mathematical reasoning (Willsey et al., 2021). In SQL query optimization, it rewrites queries into time-efficient forms (Barbulescu et al., 2024). In hardware synthesis, it supports cost-aware rewrites such as optimized matrix multiplication (Ustun et al., 2022). In formal methods, it accelerates automated theorem proving through normalization and equivalence checking (Kurashige et al., 2024). In mathematical reasoning, it is used to generate paraphrases of math expressions (Zheng et al., 2025).

In symbolic regression, de França & Kronberger (2023) leverages e-graphs to mitigate overparameterization in candidate expressions. de França & Kronberger (2025); de Franca & Kronberger (2025) further incorporates e-graphs into genetic programming (GP) to detect and eliminate redundant individuals, encouraging GP to explore novel expressions. A recent follow-up work (de França & Kronberger, 2025) provides a richer API for interacting with GP. This study advances existing work by offering a unified interface for encoding known mathematical equalities as e-graphs, enabling equivalence-aware learning across several modern symbolic regression algorithms, together with theoretical guarantees.

**Equivalence-aware Learning.** Equivalence and symmetry have long been recognized as crucial for improving efficiency in search and learning. In MCTS, transposition tables exploit equivalence by merging nodes that represent the same underlying state, avoiding redundant rollouts and accelerating convergence (Childs et al., 2008). More recent extensions explicitly leverage symmetries to prune symmetric branches of the search space (Saffidine et al., 2012; Leurent & Maillard, 2020). In reinforcement learning, symmetries in the state–action space have been used to accelerate convergence (Grimm et al., 2020). LLMs also benefit from equivalence-awareness, particularly in code generation (Sharma & David, 2025).

# 5 EXPERIMENTS

We show that (1) EGG enhances existing learning algorithms in discovering expressions with smaller Normalized MSEs. (2) In case studies, EGG consistently exhibits both time and space efficiency. The detailed experimental setups and datasets used in each comparison, are provided in Appendix C.

## 5.1 OVERALL BENCHMARKS

**Impact of EGG on MCTS.** We conduct two analyses to evaluate the impact of integrating EGG into standard MCTS: (1) the median normalized MSE of the TopK ($K = 10$) expressions identified at the end of training, and (2) The growth of the search tree, measured by the number of explored nodes over learning iterations.

Table 1 shows that EGG-MCTS consistently discovers expressions with lower normalized quantile scores compared to standard MCTS. The dataset is selected from Jiang & Xue (2023) as the expressions contain $\sin, \cos$ operators, which contain many symbolic-equivalence variants.

Figure 3(Left) illustrates that EGG-MCTS maintains a broader and deeper search tree, indicating exploration of a larger and more diverse search space. Across various datasets, augmenting MCTS with

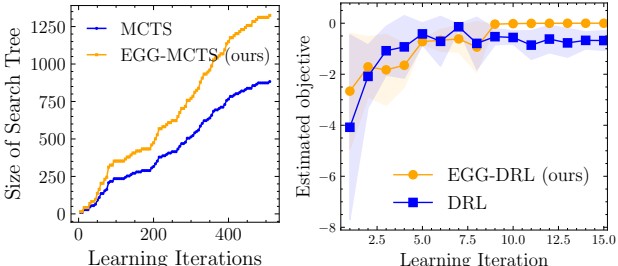

Figure 3: On the "sincos(3,2,2)" dataset, we show **(left)** search tree size over learning iterations for MCTS and EGG-MCTS, and also **(right)** empirical mean and standard deviation of the estimated quantity for DRL and EGG-DRL.

Table 1: On Trigonometric datasets, median NMSE values of the best-predicted expressions found by all the algorithms. The 3-tuples at the top $(\cdot, \cdot, \cdot)$ indicate the number of free variables, singular terms, and cross terms in the ground-truth expressions generating the dataset. The set of operators is $\{\sin, \cos, +, -, \times\}$. The best result in each column is underlined.

| | Noiseless Setting | | | | Noisy Setting | | | |
|---|---|---|---|---|---|---|---|---|
| | $(2,1,1)$ | $(3,2,2)$ | $(4,4,6)$ | $(5,5,5)$ | $(2,1,1)$ | $(3,2,2)$ | $(4,4,6)$ | $(5,5,5)$ |
| EGG-MTCS | <1E-6 | <1E-6 | 0.006 | 0.009 | 0.005 | 0.012 | 0.091 | 0.121 |
| MTCS | 0.006 | 0.033 | 0.144 | 0.147 | 0.015 | 0.007 | 0.138 | 0.150 |
| EGG-DRL | 0.020 | 0.161 | 2.381 | 2.168 | 0.07 | 0.35 | 5.09 | 5.67 |
| DRL | 0.030 | 0.277 | 2.990 | 2.903 | 0.09 | 0.44 | 2.46 | 14.44 |

Table 2: Comparison of LLM, and EGG-LLM models on different scientific benchmark problems measured by the NMSE metric. The best result in each column is underlined.

| | Oscillation I | | Oscillation II | | Bacterial growth | | Stress-Strain | |
|---|---|---|---|---|---|---|---|---|
| | IID↓ | OOD↓ | IID↓ | OOD↓ | IID↓ | OOD↓ | IID↓ | OOD↓ |
| EGG-LLM (GPT3.5) | <1E-6 | 0.0004 | <1E-6 | <1E-6 | 0.0121 | 0.0198 | 0.0202 | 0.0419 |
| LLM-SR (GPT-3.5) | <1E-6 | 0.0005 | <1E-6 | 3.81E-5 | 0.0214 | 0.0264 | 0.0210 | 0.0516 |
| EGG-LLM (Mistral) | <1E-6 | 0.0002 | 0.0021 | 0.0114 | 0.0101 | 0.0107 | 0.0133 | 0.0754 |
| LLM-SR (Mistral) | <1E-6 | 0.0002 | 0.0030 | 0.0291 | 0.0026 | 0.0037 | 0.0162 | 0.0946 |

EGG improves symbolic expression accuracy. This improvement is primarily due to the effectiveness of our rewrite rules, which cover a rich set of trigonometric identities and enable efficient exploration of symbolic variants in trigonometric expression spaces.

**Impact of EGG on DRL.** Table 1 reports the median NMSE values of the best-predicted expressions discovered by EGG-DRL and standard DRL, under identical experiment settings. Expressions returned by EGG-DRL achieve a smaller NMSE value on noiseless and noisy settings. It shows that embedding EGG into DRL helps to discover expressions with better NMSE. In Figure 3 (Right), we plot the estimated objective, defined as $R(\tau_i) \log p_\theta(\tau_i)$ where each trajectory $\tau_i$ is sampled from the sequential decoder with probability $p_\theta(\tau_i)$ (see Equation 3). We plot the empirical mean and standard deviation of this objective over training iterations. The observed reduction in variance is primarily due to the symbolic variants generated via the e-graph, which enable averaging over multiple equivalent expressions and thus yield more stable gradients.

**Impact of EGG on LLM.** Following the dataset and experimental setup from the original paper (Shojaee et al., 2025), we summarize the results in Table 2. The result of LLM-SR directly uses the reported result in Shojaee et al. (2025). The results show that integrating EGG enables the LLM to discover higher-quality expressions under the same experimental conditions, as with richer feedback prompts that incorporate equivalent expressions generated by EGG.

## 5.2 CASE ANALYSIS

**Space Efficiency of EGG.** We evaluate the space efficiency of the e-graph representation in comparison to a traditional array-based approach. We benchmark the memory consumption of storing all equivalent variants of input expressions under two settings: (1) $\phi = \log(x_1 \times \ldots \times x_n)$, using the logarithmic identity $\log(ab) \rightsquigarrow \log a + \log b$, and (2) $\phi = \sin(x_1 + \ldots + x_n)$, using the trigonometric identity $\sin(a + b) \rightsquigarrow \sin(a)\cos(b) + \sin(b)\cos(a)$. Both settings yield $2^{n-1}$ equivalent variants. The array-based method explicitly stores each expression variant as a unique sequence, leading to exponential memory growth. In contrast, the e-graph compactly encodes multiple equivalent expressions by sharing common sub-expressions.

Figure 4 reports memory consumption as a function of the number of variables $n$. The results show that e-graphs use substantially less memory than the array-based representation. We also provide additional visualizations of the constructed e-graphs for $n = 2, 3, 4$: case (1) in Appendix Figure 14 and case (2) in Appendix Figure 15. It visualizes two representative e-graphs, illustrating how shared sub-expressions are stored once and reused across many variants, which underlies the space efficiency of EGG.

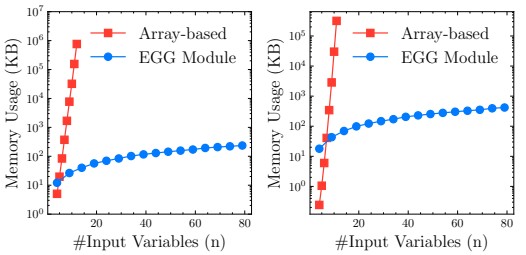

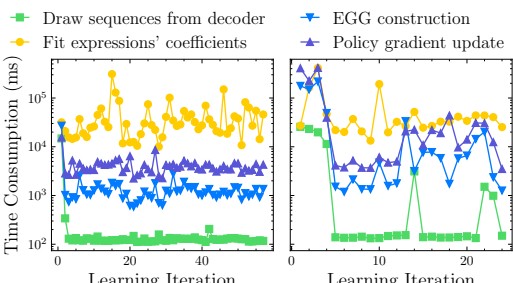

Figure 4: EGG uses less memory than the array-based approach for two settings: **(Left)** $\log\left(x_1 \times \ldots \times x_n\right)$ rewritten using $\log(ab) \rightsquigarrow \log a + \log b$. **(Right)** $\sin\left(x_1 + \ldots + x_n\right)$ rewritten using $\sin(a + b) \rightsquigarrow \sin a \cos b + \sin a \cos b$.

Figure 5: The EGG module is time efficient and introduces negligible time overhead, compared with four main computations in DRL. **Left:** LSTM. **Right:** Decoder-only Transformer.

**Time Efficiency of EGG with DRL.** As shown in Figure 5, we benchmark the runtime of the four main computations in EGG-DRL on the selected "sincos(3,2,2)" dataset: (1) sampling sequences of rules from the sequential decoder, (2) fitting coefficients in symbolic expressions to the training data, (3) generating equivalent expressions via EGG, and (4) computing the loss, gradients, and updating the neural network parameters. We consider two settings for the sequential decoder: a 3-layer LSTM with hidden dimension 128, and a decoder-only Transformer with 6 attention heads and hidden dimension 128.

The EGG module contributes minimal computational overhead relative to more expensive steps such as coefficient fitting and neural network parameter updates. The runtime of EGG depends on the size of the rewrite-rule set. As more rules are included, the e-graphs maintain increasingly large sets of equivalent expressions. This highlights the practicality of incorporating EGG into DRL-based symbolic regression frameworks.

**Additional Visualizations of E-graph Construction.** To further demonstrate the effectiveness of the proposed EGG module, we present additional visualizations of e-graph construction generated with our API on 7 selected complex expressions from the Feynman dataset (Udrescu & Tegmark, 2020). Each visualization highlights a different set of rewrite rules (see Appendix D.2). These examples further illustrate that EGG can simplify and transform complex scientific expressions in practical settings.

## 6 CONCLUSION

In this paper, we introduced EGG-SR, a unified framework that integrates symbolic equivalence into symbolic regression through equality graphs (e-graphs) to accelerate the discovery of optimal expressions. Our theoretical analysis establishes the advantages of EGG-MCTS over standard MCTS and EGG-DRL over conventional DRL algorithms. Extensive experiments further demonstrate that EGG consistently enhances the ability of existing methods to uncover high-quality governing equations from experimental data. Currently, many scientific publications use GP-based symbolic regression due to its ease of use. In future work, we plan to extend our more sophisticated solver, EGG-SR, to scientifically grounded problem settings, improving the community's computational toolkit.

**Ethics Statement.** All authors have read and commit to adhering to the ICLR Code of Ethics. This work uses only publicly available datasets and open-source models, and does not involve human subjects or human subjects data.

**Reproducibility Statement.** Appendix B describes the proposed EGG module, and the Appendix A.3 and A.2 include detailed proofs of theoretical justification. Appendix C gives the experimental setting. Appendix D collects extra experimental results.

**Acknowledgements.** We thank the reviewers for their constructive feedback, as well as Fabricio Olivetti de França for his public comments. This research was supported by TACC (CCR25054) and the U.S. Department of Energy, Office of Fusion Energy Sciences (DE-SC0024583).

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

CONTENTS

**Use of Large Language Models (LLMs).** A large language model (LLM) was used exclusively for language refinement, including improvements in grammar, clarity, and readability. All research ideas, methodology, analyses, and conclusions are entirely the authors' own. The authors have thoroughly reviewed the final manuscript and accept full responsibility for its content.

**Availability of EGG, Baselines and Dataset.** Please find our code repository at:

```
https://github.com/jiangnanhugo/egg-sr
```

- **EGG**. The implementation of our EGG module is located in the folder `src/equality_graph/`.

- **Datasets**. The list of datasets used in our experiments can be found in `datasets/scibench/scibench/data/`.

- **Baselines**. The implementations of several baseline algorithms, along with our adapted versions, are provided in the `algorithms/` folder. Execution scripts for running each algorithm are also included.

We also provide a `README.md` file with instructions for installing the required Python packages and dependencies.

We summarize the supplementary material as follows: Section B provides implementation details of the proposed EGG module. Sections A.2 and A.3 present a detailed theoretical analysis of the proposed method. Section C describes the experimental setup and configurations. Finally, Section D presents additional experimental results.

# A  THEORETICAL JUSTIFICATION

## A.1  NECESSARY DEFINITIONS OF MARKOV DECISION PROCESS.

Consider a deterministic, finite-horizon Markov Decision Process (MDP) with state space $\mathcal{S}$ and action space $\mathcal{A}$. At each stage $t \in \{0, 1, \ldots, H - 1\}$ the agent observes its current state $s_t \in \mathcal{S}$, selects an action $a_t \in \mathcal{A}$, and deterministically transitions to the next state $s_{t+1} = f(s_t, a_t)$ while receiving a bounded reward $r_t \in [0, 1]$. Let $H$ denote the maximum planning horizon. The total discounted reward over the $H$-step horizon is $\sum_{t=0}^{H-1} \gamma^t r_t$, where $\gamma \in (0, 1)$ is the discount factor. For any state–action pair $(s, a)$, the *state–action value* is defined as:

$$Q(s, a) := \max_\pi \mathbb{E}_{\tau \sim \pi} \left[ \sum_{t=0}^{H-1} \gamma^t r_t \middle| s_0 = s, \, a_0 = a \right]$$

where $\pi$ is any policy and $\tau = (s_0, a_0, \ldots, s_{H-1}, a_{H-1})$ is a trajectory generated by following $\pi$. Because the MDP is deterministic, the expectation is taken only over the agent's policy-induced action sequence. The *optimal value* of a state $s$ is:

$$V(s) := \max_{a \in \mathcal{A}} Q(s, a).$$

**Extensions to Symbolic Regression.** In our symbolic regression setting, a state corresponds to a sequence of production rules that define a partially constructed expression, or equivalently, to the class of mathematical expressions that can be generated by completing this partially-complete expression. An action represents the selection of an available production rule that extends the current expression. The state space is the set of all possible expressions of maximum length $H$, and the action space is the set of production rules.

We further provide concrete instantiations of this MDP for our Monte Carlo tree search implementation in Appendix B.4.1 and for deep reinforcement learning in Appendix B.4.2.

## A.2 Proof of Theorem 3.1

### A.2.1 Problem settings

A *planning algorithm* is any procedure that, given a model of the MDP and a limited computational budget, uses simulated rollouts to estimate the value of available actions and to recommend a single action for execution. Suppose that after $n$ simulations of a planning algorithm, the agent recommends an action $a_n$ to execute at the current state $s$. Its estimated value is $Q(s, a_n)$, obtained by evaluating the discounted return that results from first taking $a_n$ and then following an optimal policy for the remaining steps. The *simple regret* of this recommendation is

$$\texttt{regret}(n) := V(s) - Q(s, a_n), \tag{5}$$

which quantifies the *loss in discounted return* incurred by choosing $a_n$ instead of the truly optimal action. A smaller $\texttt{regret}(n)$ indicates that the planning algorithm used its limited simulation Budget more effectively to identify a near-optimal action.

Planning involves selecting promising transitions to simulate at each iteration, in order to recommend actions that minimize regret $\texttt{regret}(n)$. A popular strategy is the *optimism in the face of uncertainty* (OFU) principle, which explores actions by maximizing an upper confidence bound on the value function $V$.

In the context of symbolic regression, the planning task is therefore to sequentially apply production rules so as to construct a complete expression whose predicted outputs best match the target data, while respecting the grammar and structural constraints of the search space.

**Definition 1** (Difficulty measure). The *near-optimal branching factor* $\kappa$ of an MDP is defined as

$$\kappa := \limsup_{h \to \infty} |T_h^\infty|^{1/h} \in [1, K], \tag{6}$$

where $T_h^\infty := \left\{ a \in \mathcal{A}^h : V^* - V(a) \leq \frac{\gamma^h}{1-\gamma} \right\}$ is the set of near-optimal nodes at depth $h$.

In standard MCTS, the root of the search tree corresponds to the initial state $s_0 \in S$. A node at depth $h$ represents an action sequence $a = (a_1, \ldots, a_h) \in \mathcal{A}^h$, which leads to a terminal state $s(a)$. The search tree at iteration $n$ is denoted $\texttt{tree}(T_n)$, and its maximum expanded depth is $d_n$.

Historical research has shown that the UCT principle suffers from a theoretical worst case in which it is difficult to derive meaningful regret bounds, making it unsuitable for analyzing the benefit of EGG over standard MCTS. Therefore, we assume that MCTS operates under the Optimistic Planning for Deterministic Systems (OPD) framework (Leurent & Maillard, 2020) rather than the UCT principle.

### A.2.2 Regret bound of MCTS

**Theorem A.1** (Regret bound of MCTS (Munos (2014), chapter 5)). Let $\texttt{regret}(n)$ denote the simple regret after $n$ iterations of OPD used in MCTS. Then, we have:

$$\texttt{regret}(n) = \widetilde{\mathcal{O}}\left( n^{-\frac{\log(1/\gamma)}{\log \kappa}} \right). \tag{7}$$

Here, the soft-O version of big-O notation for time complexity is defined as: $f_n = \widetilde{\mathcal{O}}(n^{-\alpha})$ means that decays at least as fast as $n^{-\alpha}$, up to a polylog factor.

When $\kappa$ is small, only a limited number of nodes must be explored at each depth, allowing the algorithm to plan deeper, given a budget of $n$ simulations. The quantity $\kappa$ represents the branching factor of the subtree of near-optimal nodes that can be sampled by the OPD algorithm, serving as an *effective* branching factor in contrast to the true branching factor $K$. Hence, $\kappa$ directly governs the achievable simple regret of OPD (and its variants) on a given MDP.

The theoretical analysis of EGG-MCTS is built on top of the analysis procedure in Leurent & Maillard (2020). Unlike their analysis, which requires unrolling the graph into a tree and addressing potentially infinite-length paths induced by cycles, our analysis starts from the unrolled tree derived from the graph in Leurent & Maillard (2020).

**Definition 2** (Upper and Lower bounds of the Value function). Let $\texttt{tree}(T)$ be the search tree after $n$ expansions. Each node corresponds to an action sequence $(a_0, a_1, \ldots, a_{H-1})$ from the root, and let $s$ denote the MDP state reached by executing this sequence from the root state $s_0$. The true value associated with this node is the optimal value of the reached state, denoted by $V(s)$. A pair of functions $(L_n, U_n)$ is said to provide *lower and upper bounds* for $V$ on $\texttt{tree}(T)$ if

$$L_n \leq V(s) \leq U_n, \qquad \text{for internal node } s \text{ in the } \texttt{tree}(T)$$

**Definition 3** (Finer Difficulty Measure). We define the *near-optimal branching factor* associated with bounds $(L_n, U_n)$ as

$$\kappa(L_n, U_n) := \limsup_{h \to \infty} \left| T_h^\infty(L_n, U_n) \right|^{1/h} \in [1, K], \tag{8}$$

where $T_h^\infty(L, U) := \left\{ a \in \mathcal{A}^h : V^* - V(a) \leq \gamma^h \big( U(a) - L(a) \big) \right\}$.

Since $(L_n, U_n)$ tighten with $n$, the sequence of bounds $(L_n, U_n)_{n \geq 0}$ is non-increasing, i.e.,

$$0 \leq \cdots \leq L_{n-1} \leq L_n \leq V \leq U_n \leq U_{n-1} \leq \cdots \leq V_{\max}$$

they will finally converges to a limit $\kappa_\infty = \lim_{n \to \infty} \kappa(L_n, U_n)$.

### A.2.3 Regret bound of EGG-MCTS

**Theorem A.2** (Regret Bound of EGG-MCTS). Let $\kappa_n = \kappa(L_n, U_n)$, and define $\kappa_\infty = \lim_{n \to \infty} \kappa(L_n, U_n) \in [1, K]$. Then EGG-MCTS achieves the regret bound

$$\texttt{regret}_{\texttt{egg}}(n) = \widetilde{\mathcal{O}} \left( n^{-\frac{\log(1/\gamma)}{\log \kappa_\infty}} \right), \qquad \text{where } \kappa_\infty \leq \kappa \tag{9}$$

*Proof.* The analysis is similar to that of Leurent & Maillard (2020, Theorem 16). In their approach, the graph is *unrolled* into a tree to leverage the analysis of Theorem A.1. This unrolled tree contains every action sequence that can be traversed in $G_n$. There would exist a path of infinite length if the graph contains cycles. The behavior of the graph $G_n$ is therefore analyzed through its corresponding unrolled tree $\texttt{UnrollTree}(T_n)$.

Our EGG-MCTS behaves almost identically to this unrolled tree, except that it contains no paths of infinite length, since our algorithm cannot empirically generate or update along an infinite action sequence. Transporting the graph-based value bounds $(L_n, U_n)$ to the unrolled tree and applying the depth–regret argument of Leurent & Maillard (2020, Appendix A.9) yields the rate in (9). The detailed derivation is therefore omitted here for brevity. $\square$

**Remark 1.** For a class of symbolic regression problems where a large set of mathematical identities is applicable, the resulting overlaps among action paths can be extensive. In this case, the effective branching factor $\kappa_\infty$ can be much smaller than the nominal branching factor $\kappa$, leading to substantially tighter regret guarantees.

### A.3 PROOF OF THEOREM 3.2

#### A.3.1 NOTATION AND ASSUMPTIONS

Let $\tau$ be a sequence of production rules sampled from a sequential decoder with probability $p_\theta(\tau)$. $\phi$ is the expression constructed by $\tau$ following grammar definition. Let $\mathcal{S}_\phi \subseteq \Pi$ be the equivalence class (under the rewrite system $\mathcal{R}$) of sequences that deterministically construct $\phi$ or can be rewritten into $\phi$ by applying rewrite rules in $\mathcal{R}$. For notation simplicity and clarity, we define the probability over the set of sequences $\mathcal{S}_\phi$

$$q_\theta(\phi) := \sum_{\tau \in \mathcal{S}_\phi} p_\theta(\tau). \tag{10}$$

#### A.3.2 PROOF OF UNBIASEDNESS

**Lemma 1** (Key identity). *For any $\phi \in \Phi$ with $q_\theta(\phi) > 0$, $\mathbb{E}_{\tau \sim p_\theta}\left[\nabla_\theta \log p_\theta(\tau)|\phi\right] = \nabla_\theta \log q_\theta(\phi)$.*

*Proof.* By the definition of conditional probability, $\mathbb{P}(\tau|\phi) = \frac{p_\theta(\tau)}{q_\theta(\phi)}$, for $\tau \in \mathcal{S}_\phi$ and 0 otherwise. Hence,

$$\mathbb{E}\left[\nabla_\theta \log p_\theta(\tau) \mid \phi\right] = \sum_{\tau \in \mathcal{S}_\phi} \nabla_\theta \log p_\theta(\tau)\, \frac{p_\theta(\tau)}{q_\theta(\phi)} = \frac{1}{q_\theta(\phi)} \sum_{\tau \in \mathcal{S}_\phi} \nabla_\theta p_\theta(\tau) = \frac{1}{q_\theta(\phi)} \nabla_\theta \sum_{\tau \in \mathcal{S}_\phi} p_\theta(\tau)$$

$$= \frac{\nabla_\theta q_\theta(\phi)}{q_\theta(\phi)}$$

$$= \nabla_\theta \log q_\theta(\phi).$$

$\square$

**Proposition 1** (Unbiasedness). $\mathbb{E}_{\tau \sim p_\theta}\left[g(\theta)\right] = \mathbb{E}_{\tau \sim p_\theta}\left[g_{\mathsf{egg}}(\theta)\right].$

*Proof.* First, for the EGG-based estimator,

$$\mathbb{E}\left[g_{\mathsf{egg}}(\theta)\right] = \sum_{\tau \in \Pi} \mathtt{reward}(\phi)\,\nabla_\theta \log q_\theta(\phi)\, p_\theta(\tau) = \sum_{\phi \in \Phi} \sum_{\tau \in \mathcal{S}_\phi} \mathtt{reward}(\phi)\,\nabla_\theta \log q_\theta(\phi)\, p_\theta(\tau)$$

$$= \sum_{\phi \in \Phi} \mathtt{reward}(\phi)\,\nabla_\theta \log q_\theta(\phi) \underbrace{\sum_{\tau \in \mathcal{S}_\phi} p_\theta(\tau)}_{= q_\theta(\phi)}$$

$$= \sum_{\phi \in \Phi} \mathtt{reward}(\phi)\,\nabla_\theta q_\theta(\phi)$$

$$= \nabla_\theta \sum_{\phi \in \Phi} \mathtt{reward}(\phi)\, q_\theta(\phi).$$

For the standard estimator,

$$\mathbb{E}\left[g(\theta)\right] = \sum_{\tau \in \Pi} \mathtt{reward}(\tau)\,\nabla_\theta \log p_\theta(\tau)\, p_\theta(\tau) = \sum_{\tau \in \Pi} \mathtt{reward}(\tau)\,\nabla_\theta p_\theta(\tau)$$

$$= \nabla_\theta \sum_{\tau \in \Pi} \mathtt{reward}(\tau)\, p_\theta(\tau)$$

$$= \nabla_\theta \sum_{\phi \in \Phi} \sum_{\tau \in \mathcal{S}_\phi} \mathtt{reward}(\phi)\, p_\theta(\tau)$$

$$= \nabla_\theta \sum_{\phi \in \Phi} \mathtt{reward}(\phi)\, q_\theta(\phi),$$

where we used class-invariance of the reward and (10). This equals the expression obtained for $g_{\mathsf{egg}}$, proving unbiasedness. $\square$

### A.3.3 Proof of Variance Reduction

Define the $\sigma$-field generated by the expression $\phi$ as $\sigma(\phi)$. Consider the random variable

$$Z := \texttt{reward}(\tau)\, \nabla_\theta \log p_\theta(\tau) = \texttt{reward}(\phi)\, \nabla_\theta \log p_\theta(\tau),$$

where the equality uses reward invariance within each $\mathcal{S}_\phi$. By Lemma 1,

$$\mathbb{E}\,[\,Z|\phi] = \texttt{reward}(\phi)\,\mathbb{E}\,[\,\nabla_\theta \log p_\theta(\tau)|\phi] = \texttt{reward}(\phi)\,\nabla_\theta \log q_\theta(\phi) = g_{\texttt{egg}}(\theta).$$

Thus, $g_{\texttt{egg}}(\theta)$ is the *Rao–Blackwellization* of $g(\theta)$ with respect to $\sigma(\phi)$ (Casella & Robert, 1996).

**Proposition 2** (Variance reduction). $\mathbb{V}\mathrm{ar}_{\tau\sim p_\theta}\big[g_{\texttt{egg}}(\theta)\big] \leq \mathbb{V}\mathrm{ar}_{\tau\sim p_\theta}\big[g(\theta)\big]$.

*Proof.* By the law of total variance,

$$\mathbb{V}\mathrm{ar}(Z) = \mathbb{V}\mathrm{ar}\big(\mathbb{E}[Z|\phi]\big) + \mathbb{E}\big[\mathbb{V}\mathrm{ar}(Z|\phi)\big] \geq \mathbb{V}\mathrm{ar}\big(\mathbb{E}[Z|\phi]\big).$$

Substituting $Z = g(\theta)$ and $\mathbb{E}[Z|\phi] = g_{\texttt{egg}}(\theta)$ yields

$$\mathbb{V}\mathrm{ar}_{\tau\sim p_\theta}\big[g_{\texttt{egg}}(\theta)\big] \leq \mathbb{V}\mathrm{ar}_{\tau\sim p_\theta}\big[g(\theta)\big]$$

$\square$

Combining Propositions 1 and 2 proves Theorem 3.2.

**Remark on baselines.** If a baseline $b(\cdot)$ independent of the sampled action (sequence) is included, both estimators remain unbiased, and the Rao–Blackwell argument applies to the centered variable as well, so the variance reduction still holds (and can be further improved by an appropriate baseline choice).

## B    IMPLEMENTATION DETAILS

### B.1    VISUALIZATION OF EXPRESSION CONSTRUCTION VIA CONTEXT-FREE GRAMMAR

Figure 6 further illustrates how an expression such as $\phi = c_1 \log(x_1)$ can be constructed from the start symbol by sequentially applying grammar rules. We begin with the multiplication rule $A \to A \times A$, which expands the initial symbol $A$ in $\phi = A$ to yield $\phi = A \times A$. Continuing this process, each non-terminal symbol is recursively expanded using the appropriate grammar rules until we obtain the valid expression $\phi = c_1 \log(x_1)$.

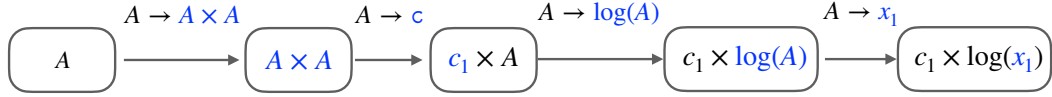

Figure 6: Transforming a sequence of grammar rules into a valid expression. At each step, the *first* non-terminal symbol inside the squared box is expanded. The expanded parts are color-highlighted.

The next step is to determine the optimal coefficient $c_1$ in the expression $\phi = c_1 \log(x_1)$. More generally, suppose the expression contains $m$ free coefficients. Given training data $D = \{(\mathbf{x}_i, y_i)\}_{i=1}^{N}$, we optimize these coefficients using a gradient-based method such as BFGS by solving

$$\mathbf{c}^* \leftarrow \arg \min_{\mathbf{c} \in \mathbb{R}^m} \frac{1}{N} \sum_{i=1}^{N} \ell(\phi(\mathbf{x}_i, \mathbf{c}), y_i),$$

where the loss function $\ell$ is typically the normalized mean squared error (NMSE), which is defined in Equation (12).

### B.2    IMPLEMENTATION OF REWRITE RULES

The following snippet defines the Python implementation of a rewrite rule from a known mathematical identity. The function `rules_to_s_expr` converts the input list representation into an internal symbolic expression tree:

```python
class Rule:
    "each rule is defined as: LHS \leadsto RHS."
    def __init__(self, lhs, rhs):
        self.lhs = rules_to_s_expr(lhs)
        self.rhs = rules_to_s_expr(rhs)
```

Each rule encodes one direction of applying a mathematical identity. Prior research (de França & Kronberger, 2025) has primarily used rewrite rules to simplify expressions, typically rewriting a longer expression into a shorter one. In contrast, our work considers both directions of each identity, enabling us to systematically generate the complete set of symbolically equivalent expressions.

We define several types of rewrite rules based on well-known mathematical identities: Let $a, b$ denote arbitrary variables, coefficients, or sub-expressions.

1. Commutative law. $a+b = b+a$ or $a*b = b*a$ will be converted into rules: $a+b \rightsquigarrow b+a, a*b \rightsquigarrow b*a$.

```python
# Commutative laws
Rule(lhs=['A->(A+A)', 'A->a', 'A->b'], rhs=['A->(A+A)', 'A->b', 'A->a'])
Rule(lhs=['A->A*A', 'A->a', 'A->b'], rhs=['A->A*A', 'A->b', 'A->a'])
```

2. Distributive laws: $(a + b)^2 \rightsquigarrow a^2 + 2ab + b^2$.

3. Factorization. Expressions can be decomposed into simpler components: $a^2 - b^2 \rightsquigarrow (a-b)(a+b)$.

4. Exponential and logarithmic identities: $\exp(a + b) \rightsquigarrow \exp(a) \times \exp(b)$, $\log(ab) \rightsquigarrow \log(a) + \log(b)$, and also $\log(a^b) \rightsquigarrow b \log a$,

Table 3: The list of rewrite rules for trigonometric functions, each derived from a known trigonometric law.

| Category | rewrite rules |
|---|---|
| Sum-to-Product Identities | $\sin(a) + \sin(b) \rightsquigarrow 2\sin\left(\frac{a+b}{2}\right)\cos\left(\frac{a-b}{2}\right)$ 
 $\cos(a) + \cos(b) \rightsquigarrow 2\cos\left(\frac{a+b}{2}\right)\cos\left(\frac{a-b}{2}\right)$ |
| Product-to-Sum Identities | $\sin(a)\cos(b) \rightsquigarrow \frac{1}{2}\left[\sin(a+b) + \sin(a-b)\right]$ 
 $\cos(a)\cos(b) \rightsquigarrow \frac{1}{2}\left[\cos(a+b) + \cos(a-b)\right]$ 
 $\sin(a)\sin(b) \rightsquigarrow \frac{1}{2}\left[\cos(a-b) - \cos(a+b)\right]$ |
| Double Angle Formulas | $\sin(a+a) \rightsquigarrow 2\sin(a)\cos(a)$ 
 $\cos(a+a) \rightsquigarrow \cos^2(a) - \sin^2(a)$ 
 $\cos(a+a) \rightsquigarrow 2\cos^2(a) - 1$ 
 $\cos(a+a) \rightsquigarrow 1 - 2\sin^2(a)$ 
 $\tan(a+a) \rightsquigarrow \frac{2\tan(a)}{1-\tan^2(a)}$ |
| Pythagorean Identities | $\sin^2(a) + \cos^2(a) \rightsquigarrow 1$ 
 $\sec^2(a) - \tan^2(a) \rightsquigarrow 1$ 
 $\csc^2(a) - \cot^2(a) \rightsquigarrow 1$ |
| Half-Angle Formulas | $\sin^2\left(\frac{a}{2}\right) \rightsquigarrow \frac{1-\cos(a)}{2}$ 
 $\cos^2\left(\frac{a}{2}\right) \rightsquigarrow \frac{1+\cos(a)}{2}$ 
 $\tan^2\left(\frac{a}{2}\right) \rightsquigarrow \frac{1-\cos(a)}{1+\cos(x)}$. |
| Sum and Difference Formulas | $\sin(a \pm b) \rightsquigarrow \sin(a)\cos(b) \pm \cos(a)\sin(b)$ 
 $\cos(a \pm b) \rightsquigarrow \cos(a)\cos(b) \mp \sin(a)\sin(b)$ 
 $\tan(a \pm b) \rightsquigarrow \left(\tan(a) \pm \tan(b)\right) / \left(1 \mp \tan(a)\tan(b)\right)$ |

5. The rewrite rules for trigonometric identities are summarized in Table 3. For example, The rule $\sin(a+b) \rightsquigarrow \sin(a)\cos(b) + \sin(b)\cos(a)$ is implemented as follow:

```
Rule(
lhs=['A->sin(A)','A->(A+A)', 'A->a', 'A->b'],
rhs=['A->(A+A)', 'A->A*A', 'A->sin(A)','A->a','A->cos(A)', 'A->b',
                 'A->cos(A)','A->a','A->sin(A)', 'A->b'])
```

A visualization of this rule being applied to an expression is shown in Figure 12.

For practical reasons, we did not introduce rewrite rules that could make the e-graph exponentially large. Rules such as $a + b - b \rightsquigarrow a - b + b$ are omitted, since they lead to infinite symbolic variants. Such extreme cases are still beyond the capability of the current EGG module.

Note that each rewrite rule is well-defined only on its feasible domain and is not applicable in out-of-domain cases. In our symbolic regression setting, such cases typically indicate that the completed expression does not fit the training data and will trigger numerical errors during evaluation. For example, the rule $\log(a \times b) \rightsquigarrow \log(a) + \log(b)$ is valid when $a, b > 0$, but is undefined when $a < 0$ or $b < 0$. The expressions $\log(x_1^3) + \log(x_2^2)$ and $\log(x_1^3 x_2^2)$ evaluated on datasets containing negative inputs will yield $-\infty$ in floating-point arithmetic, causing the NMSE evaluation to become Not-a-Number (NaN). This indicates that such candidate expressions are incompatible with the dataset. Systematically encoding and enforcing domain constraints for all rewrite rules requires a careful study of the underlying mathematical identities, and we leave this as an interesting direction for future work.

### B.3 IMPLEMENTATION OF E-GRAPH

Here we provide the implementation of the EGG data structure proposed in Section 3.1. An e-graph consists of two core components: *e-nodes* and *e-classes*. The following snippet presents the skeleton structure of these two components:

```
class ENode:
    def __init__(self, operator: str, operands: List):
```

```python
        self.operator, self.operands = operator, operands

    def canonicalize(self):
        # Convert to the canonical form of this ENode

class EClassID:
    def __init__(self, id):
        self.id = id
        self.parent = None
    def find_parent(self):
        # Return the representative parent ID (using union-find algorithm)
    def __repr__(self):
        return 'e-class{}'.format(self.id)
```

The e-graph structure is implemented in the following class. The dictionary `hashcons` stores all the edges from an e-node to an e-class. The API supports the key operations of (1) adding new expressions to the e-graph by calling `add_enode`, and (2) merging two e-classes into one by calling `merge`.

```python
class EGraph:
    # hashcons: stores a mapping from ENode to its corresponding EClass
    hashcons: Dict[ENode, EClassID] = {}

    def add_enode(self, enode: ENode):
        # Add an ENode into the e-graph

    def eclasses(self) -> Dict[EClassID, List[ENode]]:
        # Return all e-classes and their associated e-nodes

    def merge(self, a: EClassID, b: EClassID):
        # Merge two e-classes using the union-find structure

    def substitute(self, pattern: Node, ...) -> EClassID:
        # Construct a new expression in the e-graph from a rule RHS
```

### B.3.1 CONSTRUCTION

The e-graph is built by repeatedly applying a set of rewrite rules to the e-graph. Each rewrite rule consists of a left-hand side (LHS) and a right-hand side (RHS) expression. The function `apply_rewrite_rules` determines which new subgraphs to construct and where to merge them within the existing e-graph.

Specifically, the LHS pattern of each rule is matched against the e-graph using the `ematch` function, which identifies all subexpressions that match the pattern. For each match, the RHS is instantiated using the extracted variable bindings, and the resulting expression is inserted into the e-graph and merged with the corresponding e-class. The `equality_saturation` function invokes `apply_rewrite_rules` for a fixed number of iterations until saturation is reached.

```python
def apply_rewrite_rules(eg: EGraph, rules: List[Rule]):
    eclasses = eg.eclasses()
    matches = []
    for rule in rules:
        for eid, env in ematch(eclasses, rule.lhs):
            matches.append((rule, eid, env))
    for rule, eid, env in matches:
        new_eid = eg.substitute(rule.rhs, env)
        eg.merge(eid, new_eid)
    eg.rebuild()

def equality_saturation(fn, rules, max_iter=20):
    eg = EGraph(rules_to_s_expr(fn))
    for it in range(max_iter):
```

```
        apply_rewrite_rules(eg, rules)
```

We employ the Graphviz API to visualize the e-classes, e-nodes, and their connections. Several visualization example is shown in section D.

### B.3.2 EXPRESSION EXTRACTION

**Cost-based extraction** retrieves a simplified expression by minimizing a user-defined cost over operators. The procedure returns an expression tree with the lowest total cost, where the cost is computed across all operators. Each e-class is assigned a cost equal to that of its cheapest e-node, and the cost of an e-node is defined recursively as

$$\text{cost(enode)} = \text{cost(operator)} + \sum_{\text{child e-classes}} \text{cost(child)}. \tag{11}$$

The algorithm iteratively updates costs for all e-nodes until convergence, i.e., when no further changes occur. This strategy yields short, simplified expressions and has been used in prior work (de Franca & Kronberger, 2025; de França & Kronberger, 2025), which employed e-graphs to simplify overly complex expressions in genetic programming. It is consistent with the philosophical principle of Occam's razor, which favors simpler expressions over more complex ones.

**Random walk-based sampling**, which draws a batch of expressions randomly from the e-graph. Starting from an e-class with no incoming edges, we randomly select an e-node within the current e-class and transition to the e-classes connected to that e-node. This process continues until an e-node with no outgoing edges is reached. The obtained sequence of visited e-nodes corresponds to drawing a valid expression from the constructed e-graph.

### B.3.3 IMPLEMENTATION REMARK

Existing e-graph implementations are available in Rust (egg library, Willsey et al. (2021)), Haskell (Hegg or srtree library), and Julia (Metatheory.jl), or wrappers for Python (egglog-python, Shanabrook (2024); de França & Kronberger (2025)). While these frameworks offer rich APIs for diverse use cases, none is directly tailored to *grammar-based* symbolic expressions.

While packages such as `SymPy` provide APIs like `simplify` and `factor`, these functions map expressions to a fixed canonical form and do not support the richer class of rewrites and transformations. These APIs can return a smaller set of symbolically equivalent expressions than our EGG framework. Hence, they are not used in our implementation.

Our implementation builds upon an existing code snippet[1]. The most relevant prior implementation we identified is available in Haskell[2], the language in which e-graphs were originally developed. However, this choice complicates integration with Python-based environments, especially when interfacing with learning algorithms implemented in Python. A Python wrapper for Haskell egglog also exists[3], but its API is cumbersome and lacks clarity, limiting its usability for practical applications.

---

[1]`https://colab.research.google.com/drive/1tNOQijJqe5tw-Pk9iqd6HHb2abC5aRid`
[2]`https://github.com/folivetti/eggp`
[3]`https://github.com/egraphs-good/egglog-python`

### B.4  Implementation of Egg-embedded Symbolic Regression

#### B.4.1  Implementation of Egg-MCTS

**Markov Decision Process Definition for MCTS.**   We model the search process as a finite-horizon Markov decision process (MDP). A state corresponds to a partially constructed expression, and each node in the MCTS search tree represents one such state. An action corresponds to the application of a single production rule to the first non-terminal symbol in the current state. The transition dynamics are deterministic: given a state–action pair, the successor state is uniquely determined by the application of the selected production rule. Rewards are assigned only at terminal states (leaf nodes of the search tree), obtained by evaluating the completed expression on the dataset according to an evaluation metric. The episode length is thus equal to the number of applied production rules, and we set the discount factor to 1. A trajectory is a sequence of state–action transitions corresponding to a path from the root node to a leaf node in the search tree, resulting in either a valid or an invalid expression. The MDP horizon is finite: each episode terminates when the path from the root corresponds to a valid expression with no non-terminal symbols or when the maximum depth $H$ is reached.

**Example B.1.**  Figure 7 illustrates an example execution pipeline of EGG-MCTS. We focus on two distinct root-to-leaf paths in the search tree:

Path 1: $(A \to A + A,\ A \to \sin^2(x_2),\ A \to A + A,\ A \to \cos^2(x_2))$,

$$\text{Node } s_1:\ \sin^2(x_2) + \cos^2(x_2) + A.$$

Path 2: $(A \to A + A,\ A \to A/A,\ A \to x_1,\ A \to x_1)$,

$$\text{Node } s_2:\ x_1/x_1 + A.$$

Under the grammar in Section 2, $s_1$ corresponds to the partially specified expression $\sin^2(x_2) + \cos^2(x_2) + A$, whereas $s_2$ corresponds to $x_1/x_1 + A$. The two nodes are symbolically equivalent under the rewrite rules $\{\sin^2(a) + \cos^2(a) \rightsquigarrow 1,\ a/a \rightsquigarrow 1\}$, and both reduce to the same partially specified form $1 + A$.

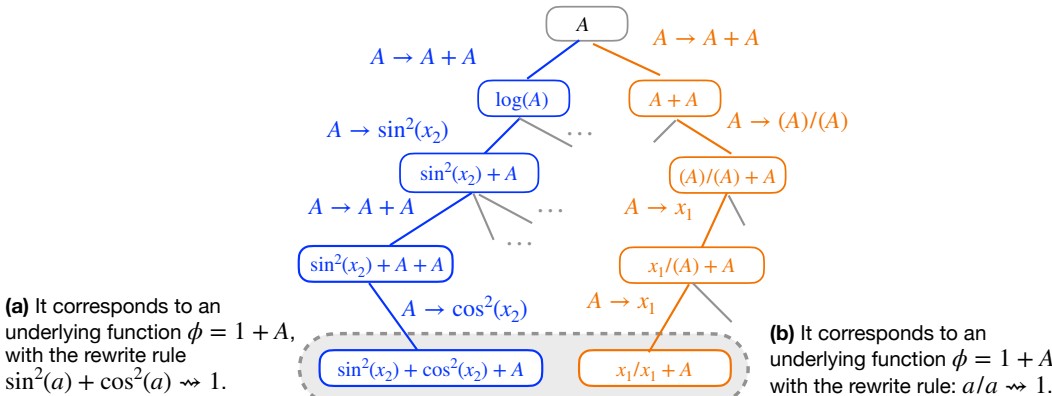

Figure 7: Two distinct paths reach leaf nodes that represent the same underlying function $\phi = 1 + A$, where $A$ denotes an arbitrary sub-expression. This equivalence follows from the rewrite rules $\sin^2(a) + \cos^2(a) \rightsquigarrow 1$ and $a/a \rightsquigarrow 1$.

Consequently, their rewards—which estimate the average goodness-of-fit on the dataset—should be approximately identical. Exploring the subtrees rooted at $s_1$ and $s_2$ independently, therefore, incurs redundant computation. In contrast, EGG identifies their equivalence and jointly updates the visit counts and reward estimates along the two paths (blue and orange), eliminating the need to further explore duplicate subtrees in subsequent iterations.

**Connection to equivalence-aware learning in MCTS.**  In many applications, multiple action sequences may lead to the same state, resulting in duplicate nodes within the search tree. To mitigate this redundancy, Saffidine et al. (2012) proposed the use of a transposition table in the context of Go, enabling the reuse of information from previous updates. Similarly, Leurent & Maillard (2020)

introduced *Monte Carlo Graph Search* (MCGS), which replaces the search tree with a graph that merges identical states. In MCGS, nodes correspond to unique states $s \in \mathcal{S}$ and edges represent state transitions, with the root node corresponding to the initial state $s_0$. At iteration $n$, if executing action $a$ from state $s$ leads to a successor state $s'$ that already exists in the graph, no new node is created.

Our EGG-MCTS adopts the same principle of detecting identical states during search, but avoids explicit graph construction. The constructed e-graph serves as a transposition table that stores and detects identical states. Following the same update rule used in a transposition table, EGG-MCTS simultaneously updates all paths in the tree that lead to the selected state. This design preserves compatibility with standard MCTS implementations while achieving a theoretical acceleration comparable to that of Leurent & Maillard (2020).

### B.4.2 IMPLEMENTATION OF EGG-DRL

**Markov Decision Process Formulation for DRL.** We model this problem as an RL agent that searches for the optimal sequence of grammar rules. At decoding step $t$, the agent samples a rule $\tau_t$ conditioned on the previously selected rules $\tau_1, \ldots, \tau_{t-1}$. We define the *state* at step $t$ as the prefix of sampled rules $s_t := (\tau_1, \ldots, \tau_{t-1})$, and the grammar rules in the output vocabulary constitute the *action space* for the RL agent. The sequential decoder, parameterized by $\theta$, defines a stochastic policy

$$\pi_\theta(a_t = a'|s_t) := p_\theta(\tau_t = a'|\tau_1, \ldots, \tau_{t-1}).$$

The environment transition is deterministic: applying action $\tau_t$ in state $s_t$ yields the next state $s_{t+1} = (\tau_1, \ldots, \tau_t)$. Rewards are obtained by evaluating the completed expression on the dataset according to a chosen evaluation metric (e.g., $1/(1 + \text{NMSE}(\phi))$). The MDP has a finite horizon: an episode terminates once the output sequence corresponds to a valid expression with no non-terminal symbols, or when the maximum sequence length $H$ is reached. The objective of the RL agent is to learn a policy that selects sequences of grammar rules so as to maximize the expected reward.

**Configuration of Sequential Decoder.** The sequential decoder can be implemented using various architectures, such as RNNs (Salehinejad et al., 2017), GRUs (Chung et al., 2014), LSTMs (Greff et al., 2016), or decoder-only Transformers (Vaswani et al., 2017). The input and output vocabularies for the decoder consist of grammar rules that encode variables, coefficients, and mathematical operators. Figure 8(a) illustrates an example of the output vocabulary.

At each time step, the model predicts a categorical distribution over the next rule, conditioned on the previously generated rules. At step $t$, the decoder (denoted as "`SequentialDecoder`") takes the previously generated rules $(\tau_1, \ldots, \tau_{t-1})$ and the hidden state $\mathbf{h}_{t-1}$, and computes

$$\mathbf{h}_t = \texttt{SequentialDecoder}(\tau_{t-1}, \mathbf{h}_{t-1}),$$
$$\mathbf{z}_t = W_o \mathbf{h}_t + b_o,$$
$$p_\theta(\tau_t|\tau_1, \ldots, \tau_{t-1}) = \texttt{softmax}(\mathbf{z}_t),$$

where $W_o \in \mathbb{R}^{|V| \times d}$ is the output weight matrix, $b_o \in \mathbb{R}^{|V|}$ is the bias vector, and $|V|$ is the size of the vocabulary. The next rule $\tau_t$ is then sampled from this distribution, $\tau_t \sim p_\theta(\tau_t|\tau_1, \ldots, \tau_{t-1})$, and fed back into the decoder as input for the next step, until the sequence is completed. After $H$ steps, the full sequence $\tau = (\tau_1, \ldots, \tau_H)$ is generated with probability $p_\theta(\tau) = \prod_{t=1}^{H} p_\theta(\tau_t|\tau_1, \ldots, \tau_{t-1})$, with the convention that $\tau_1$ is a special start symbol.

Each sequence $\tau$ is then converted into an expression following the grammar definition in Section 2. If the sequence terminates prematurely, grammar rules for variables or constants are appended at random to complete the expression. Conversely, if a valid expression is obtained before the sequence ends, the remaining rules are discarded. In both cases, the probability $p_\theta(\tau)$ is updated consistently with the applied modifications.

Finally, the model parameters are updated using gradient-based optimization (e.g., the Adam optimizer) with either the classic policy gradient estimator $g(\theta)$ or the proposed EGG-based estimator $g_{\text{egg}}(\theta)$. The whole pipeline is visualized in Figure 8.

**Connection to equivalence-aware learning in DRL.** Several techniques have been introduced to reduce the variance of policy gradient estimates. One widely used approach is the control variate

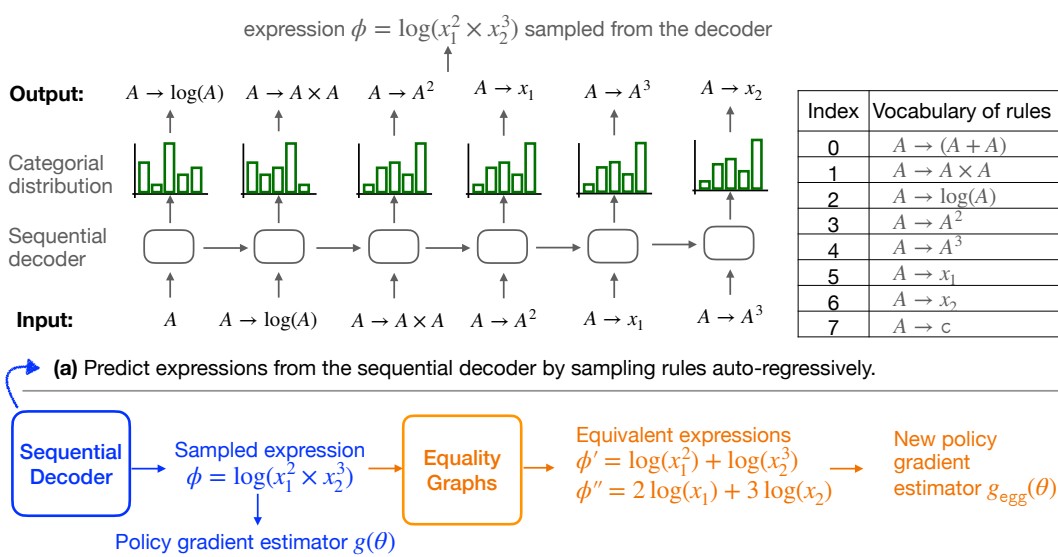

(a) Predict expressions from the sequential decoder by sampling rules auto-regressively.

(b) Learning pipeline of DRL (colored blue) and EGG-DRL(colored blue and green).

Figure 8: Framework of DRL and our EGG-DRL. **(a)** The sequential decoder autoregressively samples grammar rules according to the modeled probability distribution. **(b)** In classic DRL, the sampled expressions are directly used to compute the policy gradient estimator $g(\theta)$. While EGG-DRL constructs e-graphs from the sampled expressions, extracts symbolic variants, and computes the revised estimator $g_{\text{egg}}(\theta)$.

method, where a baseline is subtracted from the reward to stabilize the gradient (Weaver & Tao, 2001). Another idea is Rao-Blackwellization (Casella & Robert, 1996). Other approaches, such as reward reshaping (Zheng et al., 2018), modify rewards for specific state-action pairs. Inspired by stochastic variance-reduced gradient methods (Johnson & Zhang, 2013; Deng et al., 2021), Papini et al. (2018) proposed a variance-reduction technique tailored for policy gradients.

Unlike these existing methods, our proposed EGG is the first to reduce variance through the domain knowledge of the symbolic regression application, and show a seamless integration into the learning framework and attain reduced variance.

### B.4.3 IMPLEMENTATION OF EGG-LLM

Our EGG is adopted on top of the original LLM-SR (Shojaee et al., 2025).

(a) Hypothesis Generation. The LLM generates multiple candidate expressions based on a prompt describing the problem background and the definitions of each variable.

(b) Data-driven Evaluation. Each candidate expression is evaluated based on its fitness on the training dataset.

(c) **EGG-based Feedback. Inferring Equivalent Expressions.** For ease of integration with the e-graph system, we convert the generated Python functions into lambda expressions. This simplifies their manipulation and evaluation during equivalence analysis.

```python
# Original function format
def hypothesis(x1, x2, coeffs):
    expr = x1 * coeffs[0] + x2 * coeffs[0]
    return expr

# Converted into lambda format
hypothesis = lambda x1, x2, coeffs: x1 * coeffs[0] + x2 * coeffs[0]
```

This conversion allows us to transform the hypothesized Python function into an expression tree directly. For each fitted expression, we construct an e-graph and apply a predefined set of

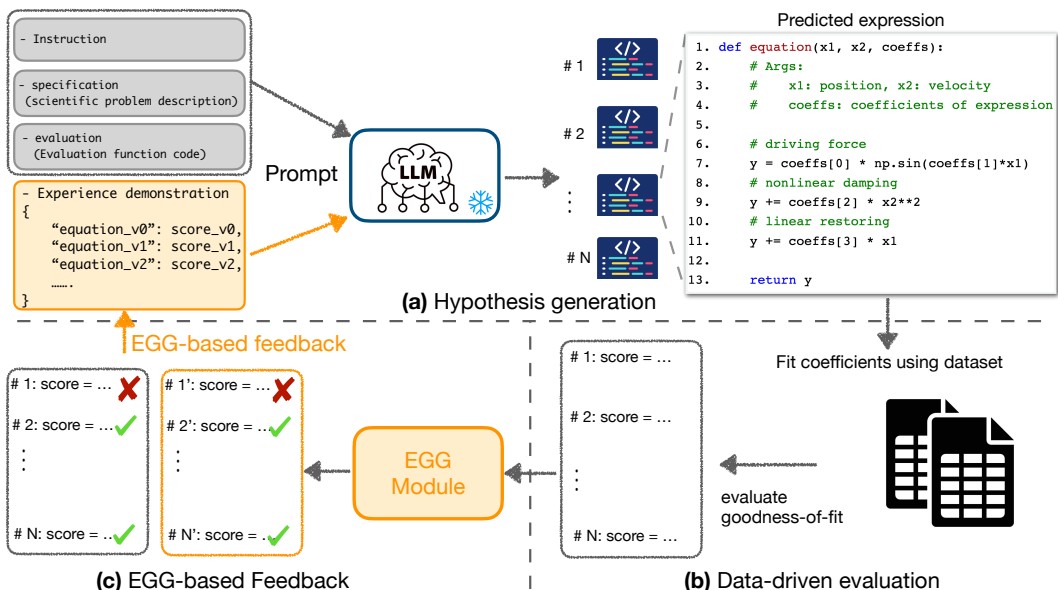

Figure 9: Pipeline of EGG-LLM. **(a)** Hypothesis Generation. The LLM receives prompts derived from the problem specification and predicts multiple candidate expressions. **(b)** Data-Driven Evaluation. The coefficients of each candidate are fitted to the dataset and then evaluated on a separate validation set to compute a goodness-of-fit score. **(c)** EGG-Based Feedback. The proposed EGG module generates symbolically equivalent variants, which are fed back into step (a) to guide the next round of hypothesis generation.

rewrite rules until reaching a fixed iteration limit. From the resulting e-graph, we sample $K$ unique equivalent expressions. Each sampled expression is then converted back into a Python function to facilitate further interaction with the LLM.

```
# obtained equivalent expression
equiv_seq = ["A->log(A)", "A->A*A", "A->x1", "A->x2"]
# Converted function format
def equiv_hypothesis(x1, x2, coeffs):
    return x1 * x2
```

In subsequent iterations, the LLM receives feedback consisting of previously generated expressions along with their fitness scores. Our feedback is enriched with both the original hypotheses and their equivalent expressions derived via EGG module, enabling the model to refine future generations. High-performing expressions are retained and further updated over multiple rounds.

### B.4.4 FINAL REMARK

The E2E-Transformer framework (Kamienny et al., 2022) is structurally very similar to the neural network used in the DRL model, with the main difference being that it is trained using a cross-entropy loss function. A straightforward way to integrate the EGG module into E2E-Transformer is through data augmentation, where additional training examples are generated using EGG.

To date, only one work has explored the use of a text-based diffusion model for symbolic regression (Bastiani et al., 2025). However, its implementation has not been released publicly. Moreover, the proposed pipeline relies on multiple interconnected components, which makes it difficult to reproduce or isolate for independent evaluation. For these reasons, we do not include a comparison between the text-based diffusion approach and EGG in this paper.

Several extensions of the baseline considered in this study offer empirical improvements but lack theoretical guarantees. For example, Tenachi et al. (2023) employ genetic programming to refine the predictions of the DRL model, and Tenachi et al. (2023) introduce unit constraints in physical

systems to eliminate infeasible expressions. Integrating EGG with these variants is an interesting direction that we leave for future investigation.

## C EXPERIMENT SETTINGS

### C.1 BASELINES AND HYPER-PARAMETERS CONFIGURATIONS

The representative baselines are selected as they all regard the search for the optimal expression as a sequential decision-making process. A detailed description of each method is provided below:

**MCTS.** Sun et al. (2023) propose to use the Monte Carlo tree search algorithm to explore the space of symbolic expressions defined by a context-free grammar (described in Section 2) to find high-performing expressions. Despite being implemented similarly, this method appears under various names in previous works (Todorovski & Dzeroski, 1997; Ganzert et al., 2010; Brence et al., 2021; Sun et al., 2023; Kamienny et al., 2023). We choose the implementation [4] from Sun et al. (2023) and serve as a representative of this family of methods.

**DRL.** Petersen et al. (2021) propose to use a recurrent neural network (RNN) trained via a (risk-seeking) reinforcement learning objective. The RNN sequentially generates candidate expressions and is optimized using a risk-seeking policy gradient to encourage the discovery of high-quality expressions. We chose this implementation [5]. We use three types of sequential decoders for the time benchmark setting. The major configurations are listed in Table 4.

Table 4: Hyperparameters for the DRL Model with different types of neural network. This configuration is also used in Figure 5.

| General Parameters | | |
|---|---|---|
| max length of output sequence | 20 | |
| batch size of generated sequence | 1024 | |
| total learning iterations | 200 | |
| Reward function | $\frac{1}{1+\text{NMSE}(\phi)}$ | |
| **Optimizer Hyperparameters** | | |
| optimizer | Adam | |
| learning rate | 0.009 | |
| entropy weight | 0.03 | |
| entropy gamma | 0.7 | |
| **Decoder-relevant Hyperparameters** | | |
| choice of decoder | LSTM | Decoder-only Transformer |
| num of layers | 3 | 3 |
| hidden size | 128 | 128 |
| dropout rate | 0.5 | / |
| number of head | / | 6 |

**LLM-SR.** Shojaee et al. (2025) proposed to use pretrained large language models, such as GPT3.5, to generate symbolic expressions based on task-specific prompts. The expressions are evaluated and iteratively refined over multiple rounds. We choose their implementation at [6].

For each baseline, we adopted the most straightforward implementation. Only minimal modifications were made to ensure that all baselines use the same input data and problem configurations, and are compatible with the proposed EGG module. We also rename the abbreviation of each method, to clearly present the core idea in each method and uniformly present the adaptation of our EGG on top of these baselines.

**Expression-related Configurations.** When fitting the values of open constants in each expression, we sample a batch of data with batch size 1024 from the data Oracle. The open constants in the expressions are fitted on the data using the BFGS optimizer[7]. We use a multi-processor library to fit multiple expressions using 8 CPU cores in parallel. This greatly reduced the total training time.

---

[4] https://github.com/isds-neu/SymbolicPhysicsLearner
[5] https://github.com/dso-org/deep-symbolic-optimization
[6] https://github.com/deep-symbolic-mathematics/LLM-SR
[7] https://docs.scipy.org/doc/scipy/reference/optimize.minimize-bfgs.html

Table 5: Hyper-parameter configurations for symbolic expression computation.

| Expressions and Dataset | |
|---|---|
| training dataset size | 2048 |
| validation and testing dataset size | 2048 |
| coefficient fitting optimizer | BFGS |
| maximum allowed coefficients | 20 |
| optimization termination criterion | error is less than $1e-6$ |

An expression containing a placeholder symbol $A$ or containing more than 20 open constants is not evaluated on the data; its fitness score is $-\infty$.

To ensure fairness, we use an interface, called `dataOracle`, which returns a batch of (noisy) observations of the ground-truth equations. To fast evaluate the obtained expression is evaluated using the Sympy library, and the step for fitting open constants in the expression with the dataset uses the optimizer provided in the Scipy library[8].

During testing, we ensure that all predicted expressions are evaluated on the same testing dataset by configuring the `dataOracle` with a fixed random seed, which guarantees that the same dataset is returned for evaluation.

We further summarize all the above necessary configurations in Table 5.

### C.2 EVALUATION METRICS

We mainly consider two evaluation criteria for the learning algorithms tested in our work: 1) the goodness-of-fit measure and 2) the total running time of the learning algorithms.

The goodness-of-fit indicates how well the learning algorithms perform in discovering unknown symbolic expressions. Given a testing dataset $D_{\text{test}} = \{(\mathbf{x}_i, y_i)\}_{i=1}^n$ generated from the ground-truth expression $\phi$, we measure the goodness-of-fit of a predicted expression $\phi_{\texttt{pred}}$, by evaluating the mean-squared-error (MSE) and normalized-mean-squared-error (NMSE):

$$\text{MSE} = \frac{1}{n}\sum_{i=1}^n (y_i - \phi_{\texttt{pred}}(\mathbf{x}_i))^2, \qquad \text{NMSE} = \frac{\frac{1}{n}\sum_{i=1}^n (y_i - \phi_{\texttt{pred}}(\mathbf{x}_i))^2}{\sigma_y^2}$$

$$\text{RMSE} = \sqrt{\frac{1}{n}\sum_{i=1}^n (y_i - \phi_{\texttt{pred}}(\mathbf{x}_i))^2}, \qquad \text{NRMSE} = \frac{1}{\sigma_y}\sqrt{\frac{1}{n}\sum_{i=1}^n (y_i - \phi_{\texttt{pred}}(\mathbf{x}_i))^2} \tag{12}$$

where the empirical variance $\sigma_y = \sqrt{\frac{1}{n}\sum_{i=1}^n \left(y_i - \frac{1}{n}\sum_{i=1}^n y_i\right)^2}$. Note that the coefficient of determination ($R^2$) metric (Nagelkerke et al., 1991; Cava et al., 2021) is equal to $(1 - \text{NMSE})$ and therefore omitted in the experiments.

---

[8] https://docs.scipy.org/doc/scipy/tutorial/optimize.html

# D EXTRA EXPERIMENTAL RESULTS

## D.1 VISUALIZATION OF EGG CONSTRUCTION

We summarize the list of example expressions, their symbolic-equivalent variants, and the visualized EGG in Table 6.

Table 6: Summary of expressions, their symbolic-equivalent variants, and the visualized o it visualization of more e-graphs obtained via EGG.

| ID | Original equation | Symbolic-equivalent variant | E-graph visualization |
|----|-------------------|-----------------------------|------------------------|
| 1 | $\log(x_1^3 x_2^2)$ | $3\log(x_1) + 2\log(x_2)$ | Figure 10 |
| 2 | $\exp(c_1 x_1 + x_2)$ | $\exp(c_1 x_1)\exp(x_2)$ | Figure 11 |
| 3 | $\sin(x_1 + x_2)$ | $\sin(x_1)\cos(x_2) + \cos(x_1)\sin(x_1)$ | Figure 12 |
| 4 | $\log(x_1 \times \ldots x_n)$ | $\log(x_1) + \ldots \log(x_n)$ | Figure 14 |
| 5 | $\sin(x_1 + \ldots + x_n)$ | omitted | Figure 15 |

**(a)** Example symbolic regressions

| ID | Original equation | Symbolic-equivalent variant | E-graph visualization |
|----|-------------------|-----------------------------|------------------------|
| I.15.3x | $\frac{x_0 - x_1 x_2}{\sqrt{c^2 - x_1^2}}$ | $\frac{x_0 - x_1 x_2}{\sqrt{c + x_1}\sqrt{c - x_1}}$ | Figure 16 |
| I.30.3 | $x_0 \frac{\sin^2(x_1 x_2/2)}{\sin^2(x_2/2)}$ | $x_0 \left(\frac{\sin(x_1 x_2/2)}{\sin(x_2/2)}\right)^2$ | Figure 17 |
| I.44.4 | $c_1 x_0 x_1 \log(x_2/x_3)$ | $c_1 x_0 x_1 (\log(x_2) - \log(x_3))$ | Figure 18 |
| I.50.26 | $x_0(\cos(x_1 x_2) + x_3 \cos^2(x_1 x_2))$ | $x_0 \cos(x_1 x_2)\left(1 + x_3 \cos(x_1 x_2)\right)$ | Figure 19 |
| II.6.15b | $\frac{3x_0}{4\pi}\frac{3\cos x_1 \sin x_1}{x_2^3}$ | $\frac{3x_0}{8\pi}\frac{\sin(2x_1)}{x_2^3}$ | Figure 20 |
| II.35.18 | $\frac{x_0}{\exp(x_1 x_2/x_3) + \exp(-x_1 x_2/x_3)}$ | $\frac{x_0}{2\cosh(x_1 x_2/x_3)}$ | Figure 21 |
| II.35.21 | $x_0 x_1 \tanh(x_1 x_2/x_3)$ | $x_0 x_1 \frac{\exp(2x_1 x_2/x_3) - 1}{\exp(2x_1 x_2/x_3) + 1}$ | Figure 22 |

**(b)** Selected complex symbolic regressions from Feynman Dataset.

**E-graph for** log **operator.** Figure 10 illustrates the e-graph construction for the expression $\log(x_1^3 x_2^2)$ using rewrite rules for log operator: $\log(a \times b) \rightsquigarrow \log(a) + \exp(b)$ and also $\log(a^b) \rightsquigarrow b \log a$. This visualization is based on the Graphviz API. We also label each e-class with a unique index for clarity. As shown in Figure 10, the saturated e-graph grows significantly larger as more rewrite rules are applied. In our full implementation, we incorporate many additional rewrite rules beyond these basic logarithmic identities, resulting in substantially larger and more complex e-graphs.

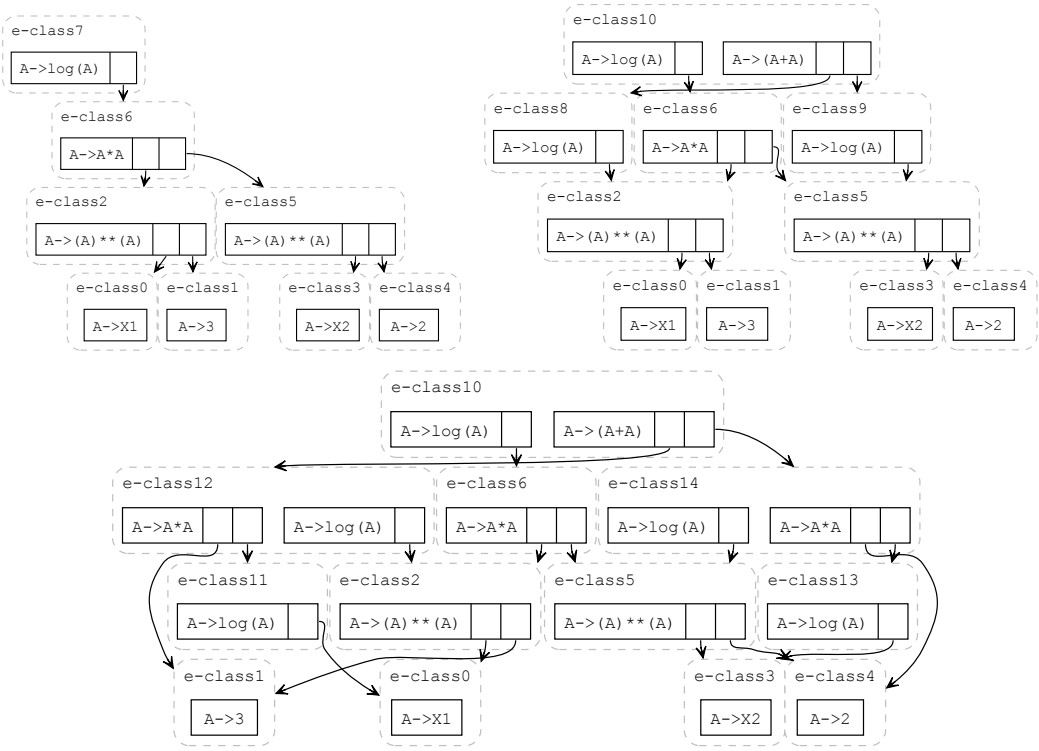

Figure 10: **(Top left)** Initialized e-graphs for expression $\log(x_1^3 x_2^2)$. **(Top Right)** The saturated e-graphs after applying one rewrite rule $\log(ab) \rightsquigarrow \log a + \log b$. **(Bottom)** The saturated e-graphs after applying two rewrite rules: $\log(ab) \rightsquigarrow \log a + \log b$ and $\log(a^b) \rightsquigarrow b \log a$.

**E-graph for** exp **operator.** Figure 11 illustrates the e-graph construction for the expression $\exp(c_1 x_1 + x_2)$ using the exponential rewrite rule $\exp(a + b) \rightsquigarrow \exp(a) \times \exp(b)$. The input expression $\exp(c_1 \times x_1 + x_2)$ is represented as a sequence of production rules: ($A \rightarrow \exp(A),\ A \rightarrow A + A,\ A \rightarrow A \times A,\ A \rightarrow c_1,\ A \rightarrow x_1,\ A \rightarrow x_2$).

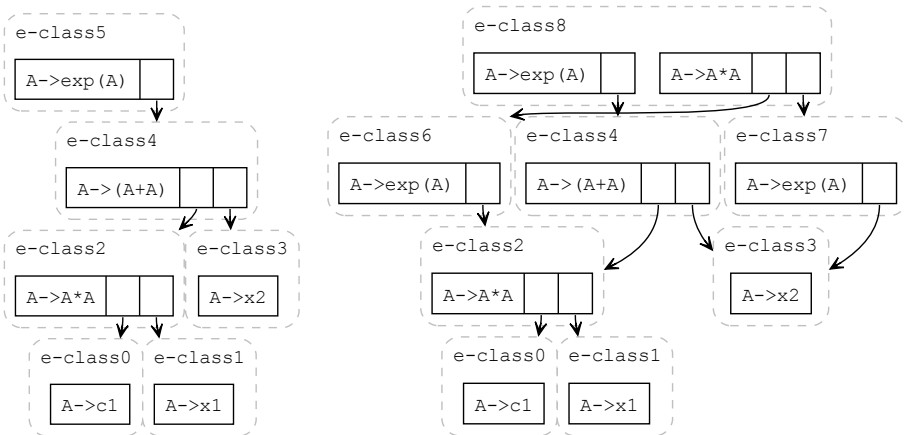

Figure 11: **(Left)** Initialized e-graph for expression $\exp(c_1 x_1 + x_2)$. **(Right)** The saturated e-graphs after applying one rewrite rule $\exp(a + b) \rightsquigarrow \exp(a)\exp(b)$.

**E-graph for** $\sin, \cos$ **operators.** Figure 12 demonstrates the application of the trigonometric identity $\sin(a + b) \rightsquigarrow \sin a \cos b + \sin b \cos a$ to the input expression $\sin(x_1 + x_2)$. The input expression is represented by the sequence of production rules $(A \to \sin(A), \ A \to A + A, \ A \to x_1, \ A \to x_2)$.

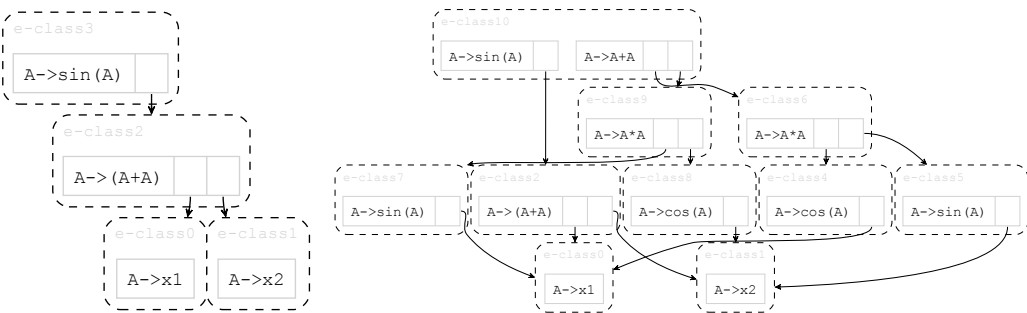

Figure 12: Applying the rewrite rule $\sin(a+b) \rightsquigarrow \sin a \cos b + \sin b \cos a$ in an e-graph representing expression $\sin(x_1 + x_2)$. **Left:** Initialized e-graph. **Right:** e-graph after applying the rewrite rule.

**E-graph for** $\partial/\partial x_i$ **operator.** Figure 13 shows the application of the partial derivative commutativity rule $\frac{\partial^2 f}{\partial x_i \partial x_j} \rightsquigarrow \frac{\partial^2 f}{\partial x_j \partial x_i}$ to the input expression $\frac{\partial^2 (x_1 + x_2)}{\partial x_1 \partial x_2}$. The derivation is represented by the following production rules: $(A \to \partial(A)/\partial x_1, \ A \to \partial(A)/\partial x_2, \ A \to x_1 + x_2)$.

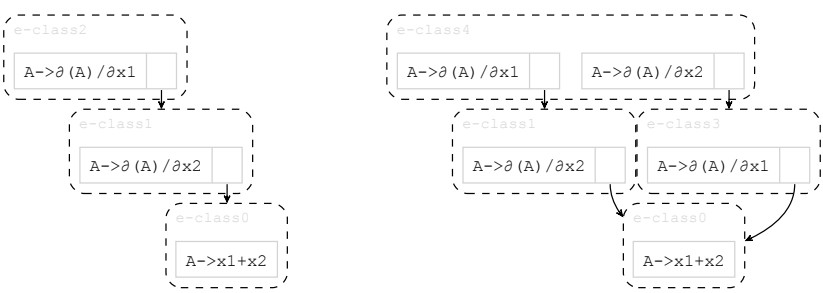

Figure 13: Applying the rewrite rule $\frac{\partial}{\partial x_i}\left(\frac{\partial f}{\partial x_j}\right) \rightsquigarrow \frac{\partial}{\partial x_j}\left(\frac{\partial f}{\partial x_i}\right)$ in an e-graph representing expression $\frac{\partial^2 (x_1 + x_2)}{\partial x_i \partial x_j}$. **Left:** Initialized e-graph. **Right:** e-graph after applying the rewrite rule.

**E-graph visualization for case analysis in section 5.2.**

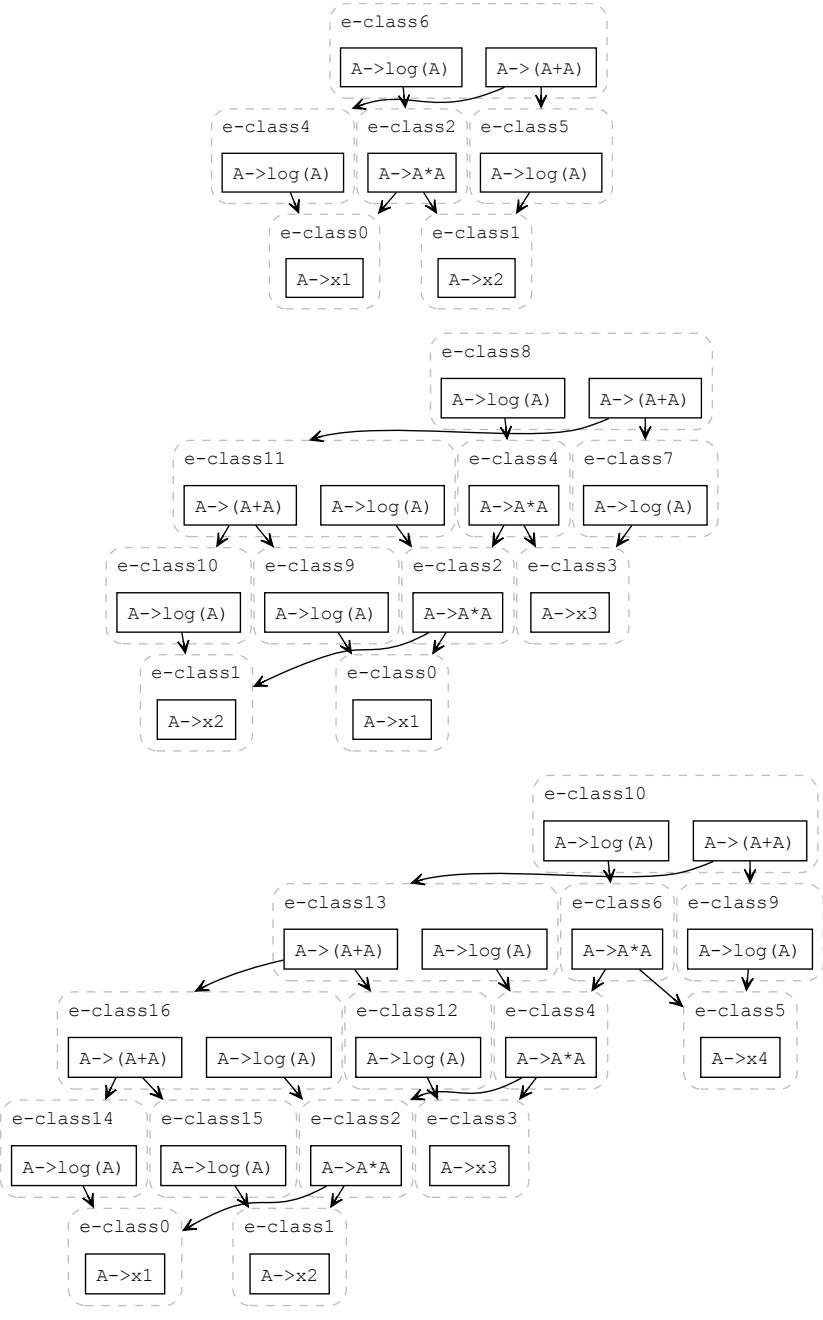

Figure 14: Visualization of e-graphs for expression $\log(x_1 \times \ldots \times x_n)$, using the rewrite rule $\log(ab) \rightsquigarrow \log a + \log b$. The three figures correspond to $n = 2, 3, 4$ accordingly.

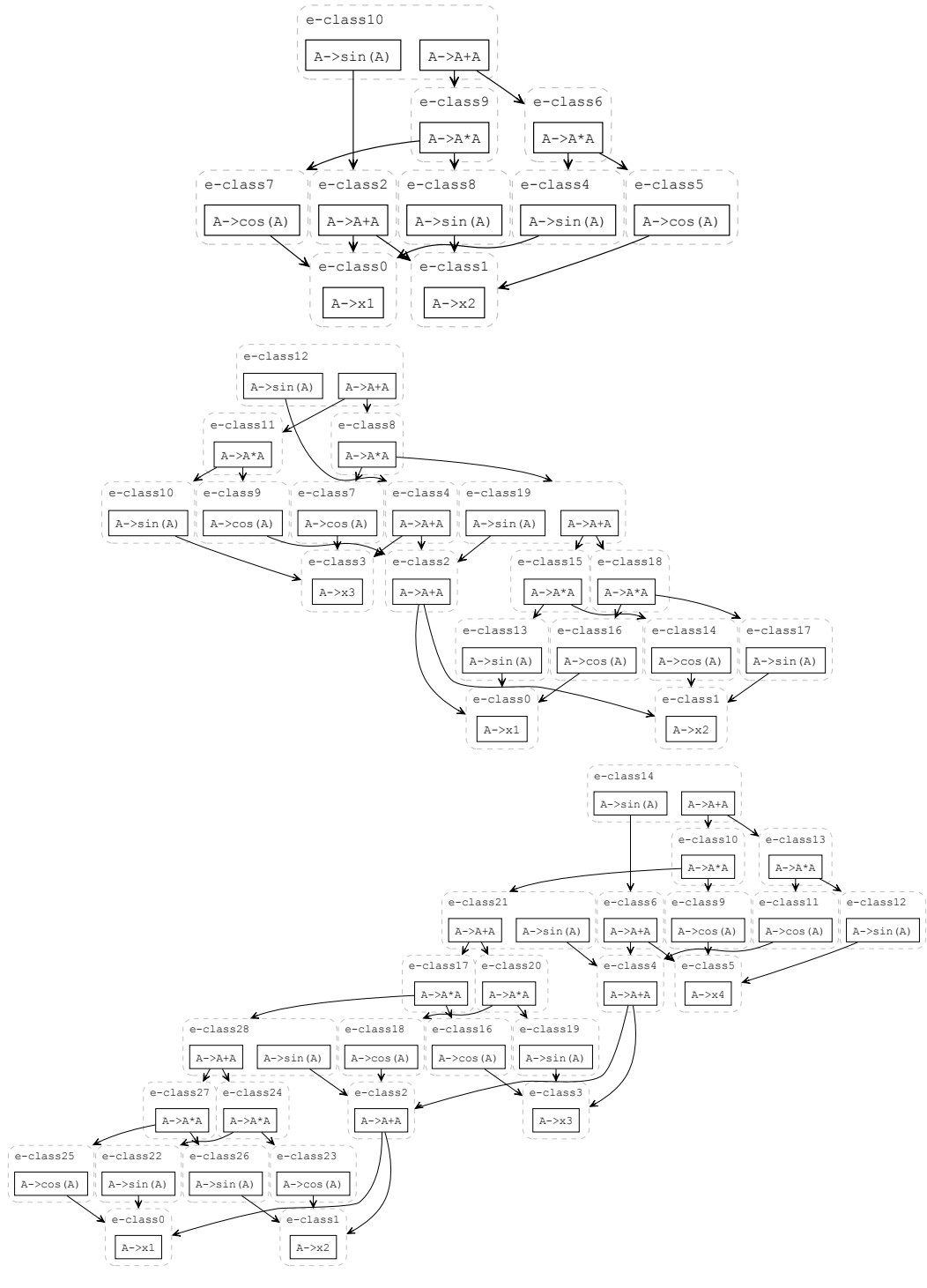

Figure 15: Visualization of e-graphs for expression $\sin(x_1 + \ldots + x_n)$, using the trigonometric rewrite rule $\sin(a + b) \rightsquigarrow \sin(a)\cos(b) + \sin(b)\cos(a)$. The three figures correspond to $n = 2, 3, 4$.

### D.2 ADDITIONAL VISUALIZATION FOR SELECTED EXPRESSIONS IN FEYNMAN DATASET

**E-graph for equation ID I.15.3x in the Feynman dataset.** Figure 16 illustrates the application of the rewrite rules $\sqrt{ab} \rightsquigarrow \sqrt{a}\sqrt{b}$ and $a^2 - b^2 = (a+b)(a-b)$ to the input expression $\frac{x_0 - x_1 x_2}{\sqrt{c_1^2 - x_1^2}}$.

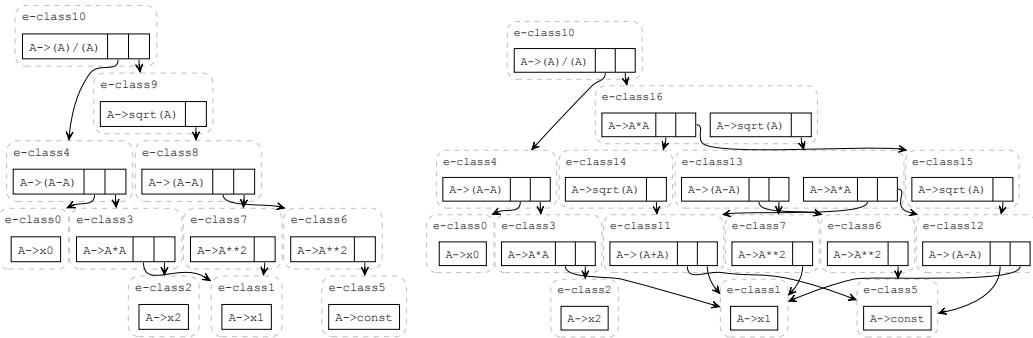

Figure 16: **(Left)** Initialized e-graph for the expression $(x_0 - x_1 x_2)/\sqrt{c_1^2 - x_1^2}$, where the equation ID is I.15.3x in the Feynman dataset. **(Right)** Saturated e-graph after applying the rewrite rule $\sqrt{ab} \rightsquigarrow \sqrt{a}\sqrt{b}$ and $a^2 - b^2 = (a+b)(a-b)$.

**E-graph for equation ID I.30.3 in the Feynman dataset.** Figure 17 illustrates the application of the rewrite rule $a^2/b^2 \rightsquigarrow (a/b)^2$ to the input expression $x_0 \frac{\sin^2(x_1 x_2/2)}{\sin^2(x_2/2)}$.

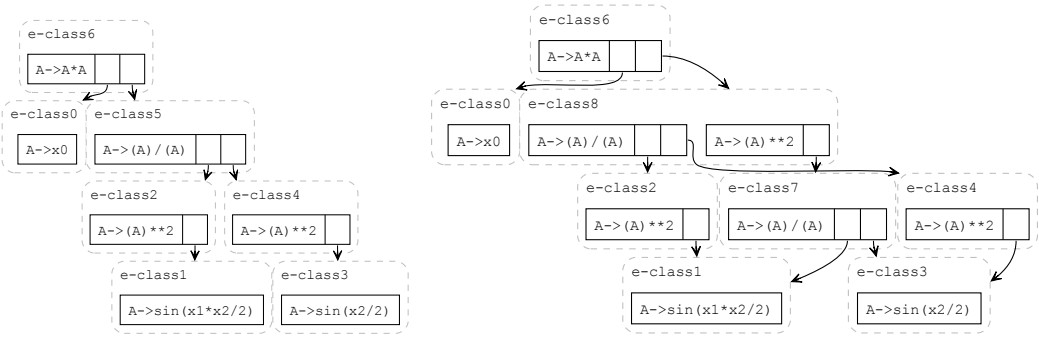

Figure 17: **(Left)** Initialized e-graph for the expression $x_0 \frac{\sin^2(x_1 x_2/2)}{\sin^2(x_2/2)}$, where the equation ID is I.30.3 in the Feynman dataset. **(Right)** Saturated e-graph after applying the rewrite rule $\log(a/b) \rightsquigarrow \log a - \log b$.

**E-graph for equation ID I.44.4 in the Feynman dataset.** Figure 18 illustrates the application of the rewrite rule $\log(a/b) \rightsquigarrow \log a - \log b$ to the input expression $c_1 x_0 x_1 \log(x_2/x_3)$.

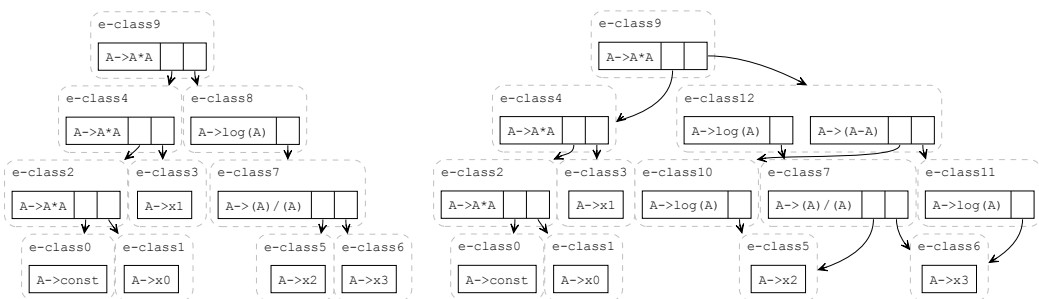

Figure 18: **(Left)** Initialized e-graph for the expression $c_1 x_0 x_1 \log(x_2/x_3)$, where the equation ID is I.44.4 in the Feynman dataset. **(Right)** Saturated e-graph after applying the rewrite rule $\log(a/b) \rightsquigarrow \log a - \log b$.

**E-graph for equation ID I.50.26 in the Feynman dataset.** Figure 19 illustrates the application of the rewrite rule $ab + ac \rightsquigarrow a(b + c)$ to the input expression $x_0(\cos(x_1 x_2) + x_3 \cos^2(x_1 x_2))$.

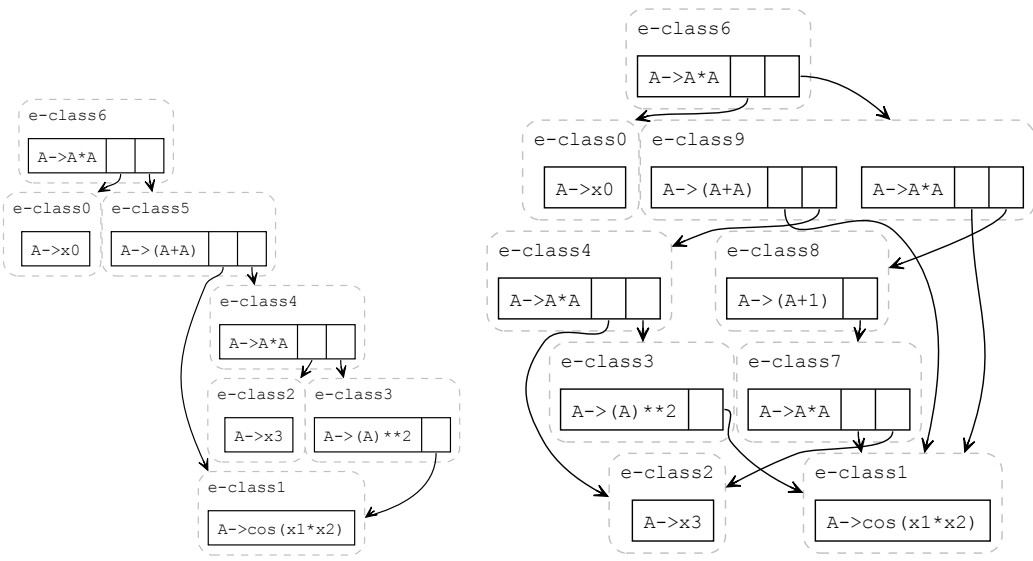

Figure 19: **(Left)** Initialized e-graph for the expression $x_0(\cos(x_1 x_2) + x_3 \cos^2(x_1 x_2))$, where the equation ID is I.50.26 in the Feynman dataset. **(Right)** Saturated e-graph after applying the rewrite rule $a + ba^2 \rightsquigarrow a(ba + 1)$.

**E-graph for equation ID II.6.15b in the Feynman dataset.** Figure 20 illustrates the application of the rewrite rule $\cos(a)\sin(a) \rightsquigarrow \frac{\sin(2a)}{2}$ to the input expression $\frac{x_0}{\exp(x_1 x_2/x_3) + \exp(-x_1 x_2/x_3)}$.

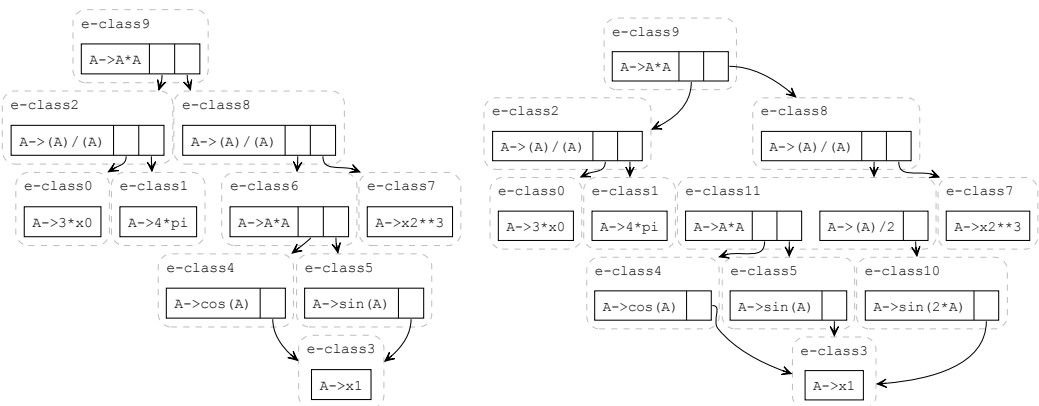

Figure 20: **(Left)** Initialized e-graph for the expression $\frac{3x_0}{4\pi}\frac{\cos x_1 \sin x_1}{x_2^3}$, where the equation ID is II.6.15b in the Feynman dataset. **(Right)** Saturated e-graph after applying the rewrite rule $\cos(a)\sin(a) \rightsquigarrow \frac{\sin(2a)}{2}$.

**E-graph for equation ID II.35.18 in the Feynman dataset.** Figure 21 illustrates the application of the rewrite rule $(\exp(a) + \exp(-a))/2 \rightsquigarrow \cosh a$ to the input expression $\frac{x_0}{\exp(x_1 x_2/x_3) + \exp(-x_1 x_2/x_3)}$.

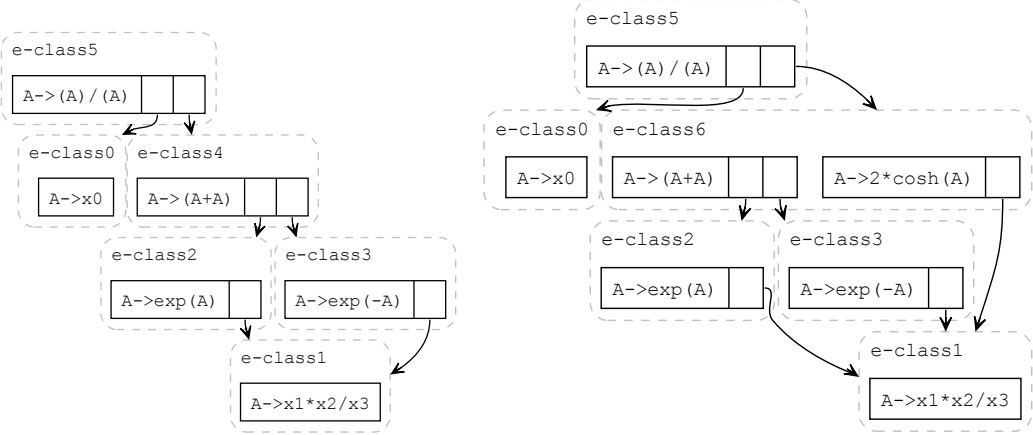

Figure 21: **(Left)** Initialized e-graph for the expression $\frac{x_0}{\exp(x_1 x_2/x_3) + \exp(-x_1 x_2/x_3)}$, where the equation ID is II.35.18 in the Feynman dataset. **(Right)** Saturated e-graph after applying the rewrite rule $(\exp(a) + \exp(-a)) \rightsquigarrow 2\cosh a$.

**E-graph for equation ID II.35.21 in the Feynman dataset.** Figure 22 illustrates the application of the rewrite rule $\tanh a \rightsquigarrow \frac{\exp(2a)-1}{\exp(2a)+1}$ to the input expression $x_0 x_1 \tanh(x_1 x_2/x_3)$.

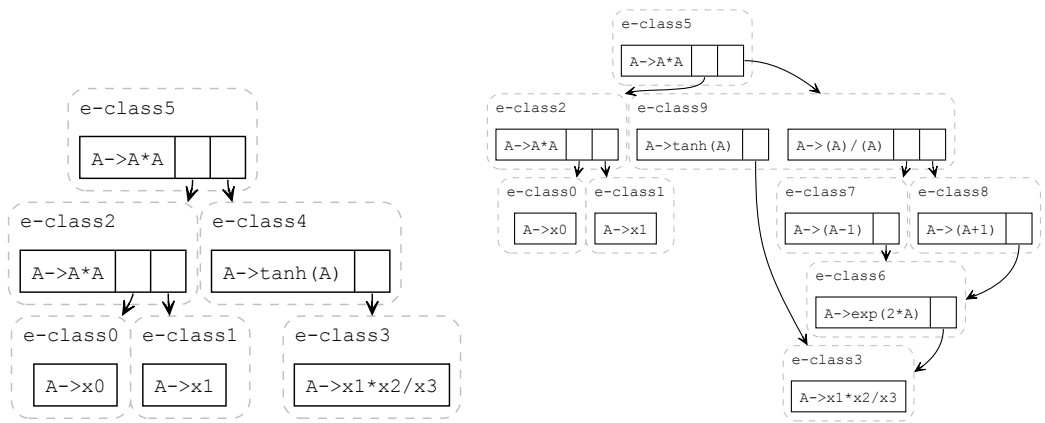

Figure 22: **(Left)** Initialized e-graph for the expression $x_0 x_1 \tanh(x_1 x_2 / x_3)$, where the equation ID is II.35.21 in the Feynman dataset. **(Right)** Saturated e-graph after applying the rewrite rule $\tanh a \rightsquigarrow \frac{\exp(2a)-1}{\exp(2a)+1}$.

