# OpenReview forum: "EGG-SR: Embedding Symbolic Equivalence into Symbolic Regression via Equality Graph"
_ICLR.cc/2026/Conference — ICLR 2026 Poster_

### Official Review · Reviewer_b8v3 · 2025-10-29

**Soundness:** 3
**Presentation:** 3
**Contribution:** 3
**Rating:** 6
**Confidence:** 5

**Summary:**

The paper introduces EGG-SR, a framework designed to improve the efficiency and effectiveness of Symbolic Regression (SR). The core problem addressed is that existing SR algorithms inefficiently explore the search space by treating syntactically different but semantically equivalent expressions as distinct candidates. EGG-SR integrates equality graphs (e-graphs) to compactly represent and manage these symbolic equivalences within a grammar-based SR setting.

**Strengths:**

- **Originality**: While e-graphs and their application to SR are not entirely new (as noted in the related works ), this paper's contribution is a novel and highly general framework. Applying the equivalence concept systematically to MCTS (search pruning), DRL (variance reduction), and LLMs (prompt enrichment) is a creative and powerful combination of ideas. The DRL variance reduction and LLM feedback mechanisms are particularly original.

- **Quality**: The paper is technically strong. The theoretical claims are well-supported by proofs in the appendix that build on established literature (e.g., Rao-Blackwellization for DRL variance ). The empirical evaluation is thorough and convincing. The authors test their claims across all three proposed algorithm variants and find consistent improvements. Crucially, they include practical analyses of memory and time overhead (Figures 4 and 5), which demonstrate the method's feasibility.

- **Clarity**: The paper is exceptionally well-written. The problem is motivated with a clear, simple example ($log(x_{1}^{2}x_{2}^{3})$) 16 that is carried through the text. The methodology is explained clearly, with helpful diagrams for the e-graph construction (Figure 1) 17, the MCTS process (Figure 2) 18, and the LLM pipeline (Figure 8)19. The distinction between the baseline algorithms and their EGG-enhanced counterparts is sharp and easy to grasp.

- **Significance**: This work addresses a fundamental and widely recognized source of inefficiency in symbolic regression. Because the EGG-SR framework is shown to be effective across diverse SR methods (search, DRL, LLM) and practical (low overhead), it has the potential for broad impact. It provides a "drop-in" enhancement that could benefit a wide range of existing and future SR systems.

**Weaknesses:**

- The process of building the e-graph, "equality saturation", involves iteratively applying all rules. While the storage is efficient (Figure 4), the construction time could become a bottleneck for very complex expressions or a very large set of rewrite rules. Figure 5  shows the overhead is negligible for one dataset, but it does not analyze how this construction time scales with expression complexity or the number of rules.
- The claim for EGG-MCTS is that it "prunes redundant subtree exploration". However, Figure 3 (Left)  shows that EGG-MCTS explores a larger search tree (more nodes) than standard MCTS. The paper states this indicates exploration of a "larger and more diverse search space", but this seems to contradict the "pruning" claim and is confusing. A clearer explanation is needed.

- The related work section (Section 4) is incomplete and overlooks several significant, recent advancements in symbolic regression. While the paper covers the main paradigms, it omits highly relevant work on generative and transformer-based SR models [2, 3, 4].
A more critical omission is the lack of any discussion or comparison with DySymNet [1]. This method is highly relevant to the present work, as DySymNet also employs a policy gradient approach (similar to EGG-DRL) to dynamically construct symbolic networks and has demonstrated strong performance, particularly on high-dimensional problems. The paper would be significantly strengthened by:

    - Including this and other recent works [2, 3, 4] to provide a more comprehensive and up-to-date overview of the field.

    - Providing at least a detailed conceptual comparison, and ideally an empirical one, between EGG-DRL and DySymNet. This is necessary to properly contextualize the DRL-based contribution of EGG-SR.

[1] Li W, Li W, Yu L, et al. A Neural-Guided Dynamic Symbolic Network for Exploring Mathematical Expressions from Data, ICML 2024

[2] Holt S, Qian Z, van der Schaar M. Deep generative symbolic regression, ICLR 2023

[3] Li W, Li W, Sun L, et al. Transformer-based model for symbolic regression via joint supervised learning, ICLR 2023

[4] Shojaee P, Meidani K, Barati Farimani A, et al. Transformer-based planning for symbolic regression, NeurIPS 2023

**Questions:**

1. The framework's success depends on the manually curated set of rewrite rules. (a) How was the rule set for the trigonometric benchmarks in Table 1  selected? Was it designed specifically for those problems? (b) How does EGG-SR's performance degrade if key rewrite rules are omitted from the set?

2. Regarding the time-efficiency in Figure 5, how does the "EGG construction" time scale with (a) the complexity (e.g., length or depth) of the initial expression and (b) the total number of rewrite rules in the system?

3. For the EGG-LLM method, how are the equivalent expressions for the feedback prompt selected from the e-graph? Do you use cost-based extraction, random sampling, or some other heuristic? How many equivalent expressions are added, and how are they formatted in the prompt to effectively guide the LLM?

4. Could you please clarify the result in Figure 3 (Left)? Why does EGG-MCTS, which is designed to avoid redundant exploration, result in a larger total search tree than standard MCTS? Does merging equivalent nodes (and sharing their visit counts/rewards) encourage a deeper, more focused search down promising paths, thus leading to a larger total node count even as redundant branches are eliminated?

I am willing to raise my score if the authors can satisfactorily address all the concerns outlined in the weaknesses and questions above.

---

> ### Author Response · Authors · 2025-11-20
> **Thanks to reviewer b8v3 for the constructive feedback on important lines of research in symbolic regression and on our experimental configuration.**
>
> We sincerely thank Reviewer b8v3 for the time and effort devoted to carefully reading our manuscript and for providing detailed, constructive feedback. Below, we address each of your concerns in turn.
>
> **1. Missing comparison with end-to-end deep symbolic approaches.**
>
> End-to-end deep symbolic regression (End2End SR) is indeed another important line of work in symbolic regression. We have added the following discussion to Section 3.3 to address this point:
>
>     > Kamienny et al. (2022) and Shojaee et al. (2023) propose to directly encode data into a Transformer and directly predict the traversal sequence of the target expression. These end-to-end models are trained using a cross-entropy loss. At inference time, they rely on pure sampling or planning-based approaches to select the best expression among multiple candidates. An intuitive way to integrate the EGG module into these frameworks is to use EGG to generate multiple correct output sequences during training. However, how to best exploit EGG during inference remains an open problem.
>
> It is now in Section 3.3 in the revised PDF:
>  https://openreview.net/pdf?id=oh9ChF7Pv0#page=7
>
> **2. Adding comparison or experiments with SymNet.**
>
> SymNet is also an important line of work in symbolic regression. We have added the following text to Section 3.3:
>
>     > Sahoo et al. (2018) and Li et al. (2024) propose representing expressions as layer-wise symbolic networks (SymNet), which are incompatible with our grammar definition. A follow-up work (Wu et al., 2024) on SymNet studies coefficient equivalence, showing that many coefficients obtained from SymNet can be merged into a single one. Systematically applying the e-graph framework to SymNet, in order to capture equivalence directly inside SymNet, remains an interesting open problem.
>
> It is now in Section 3.3 in the revised PDF:  https://openreview.net/pdf?id=oh9ChF7Pv0#page=7
>
> **3. Impact of the choice and size of rewrite rules.**
>
> For practical reasons, we did not introduce rewrite rules that could make the e-graph exponentially large. Rules such as $a + b - b \leadsto a - b + b$ are omitted, since they lead to infinite symbolic variants. Such extreme cases are still beyond the capability of the current \method module.
> As more rewrite rules are added, both the size of the e-graph and the time required to construct and maintain it increase. Designing more efficient algorithms and strategies to handle large rule sets in a systematic way is an important direction for future work.
>
> **4. Configuration of rewrite rules for Table 1.**
>
> In our experiments, we manually configured the rewrite rules used for the results in Table 1. The rule set is listed in Table 3 of the appendix, and the corresponding implementation is available at:
> https://anonymous.4open.science/r/egg-sr/src/equality_graph/equality_graph/rules.py
> It is manually constructed for general trigonometric-family expressions.
> Regarding performance: if a key rewrite rule is not included in the initial configuration, the algorithm must spend more learning iterations exploring the search space of candidate expressions before it can discover equivalent forms. In contrast, when the relevant rule is present, EGG can more quickly collapse equivalent expressions, thereby accelerating learning.
> We will present an empirical evaluation in the future revision of the paper.
>
> **5. Configuration of EGG-LLM.**
>
> For EGG-LM, we randomly sample (K = 10) sequences from the e-graph, convert each sequence into a lambda function, and then feed these as additional candidate expressions (similar in spirit to LLM-SR), together with their corresponding goodness-of-fit scores. This provides a heuristic way to inform the LLM that many distinct-looking expressions can represent the same underlying function.
>
> **6. Clarification of Figure 3 (Left): why does EGG-MCTS have a larger “search tree size”?**
>
> Thank you for raising this excellent question. In Figure 3 (Left), the “search tree size” is measured in terms of the total number of visits (i.e., how many times states are updated during backpropagation), rather than the number of distinct nodes in the tree. In EGG-MCTS, our EGG-based backpropagation updates not only the reward of the selected state but also the visit counts and rewards of all e-graph–identified equivalent states.
>
> As a result, the aggregated visit count is naturally larger than in standard MCTS, where each simulation only updates a single path.
> Intuitively, at the beginning of the search, EGG’s effect is limited because few equivalences have been discovered. As the search progresses and the e-graph identifies more and more equivalent nodes, EGG-MCTS effectively propagates information across many parts of the tree simultaneously. This leads to a large total number of visits to the search tree.

---

### Official Review · Reviewer_A6nL · 2025-10-30

**Soundness:** 3
**Presentation:** 3
**Contribution:** 4
**Rating:** 6
**Confidence:** 4

**Summary:**

This paper presents EGG-SR, a unified framework that addresses the underexplored problem of symbolic equivalence in symbolic regression (SR). The core idea is to integrate equality graphs (e-graphs) to compactly represent equivalent expressions, thereby reducing redundant search. The authors demonstrate the versatility of EGG-SR by embedding it into three distinct SR paradigms: Monte Carlo Tree Search (MCTS), Deep Reinforcement Learning (DRL), and Large Language Models (LLMs). Theoretical analysis shows tighter regret bounds for MCTS and lower gradient variance for DRL. Empirically, the EGG module consistently enhances baseline performance across various benchmarks, leading to the discovery of expressions with lower normalized mean squared error.

**Strengths:**

**Novel and Important Problem Formulation**:​​ The paper successfully identifies and tackles the fundamental issue of symbolic equivalence, a significant source of inefficiency in SR that has been largely overlooked. This focus is timely and valuable.

​**High Methodological Innovation**:​​ The primary strength is the proposal of a unified, plug-and-play framework (EGG-SR) rather than a single algorithm. Its applicability across diverse paradigms (MCTS, DRL, LLM) demonstrates remarkable generality and conceptual elegance.

**​Strong Theoretical and Empirical Validation**:​​ The work is supported by non-trivial theoretical guarantees, which substantiate the method's benefits. The experiments are comprehensive, convincingly showing performance improvements across multiple baselines and datasets. The inclusion of efficiency analyses (time/space) and rich e-graph visualizations strengthens the empirical contribution.

**Weaknesses:**

**Dependence on Manually Defined Rewrite Rules**:​​ The effectiveness of EGG-SR currently relies on a pre-defined set of rewrite rules. The framework's power and generality could be further amplified by exploring methods to learn or automatically discover these rules from data, as noted by the authors in the conclusion.

**Writings**: There is a significant distance between Figure 1 and the corresponding explanatory text (Section 3.1). To improve the reader's flow, I suggest moving the figure closer to its initial description or adding a cross-reference (e.g., "as illustrated in Figure 1") when it is first mentioned in the text. Plus, there appears to be an error in the reported minimum value for the second column under the "noisy setting" in Table 1.

**Questions:**

Concerns and questions are described in "Weaknesses".

---

> ### Author Response · Authors · 2025-11-20
> **Thanks reviewer A6nL for the insightful feedback on the Dependence on Manually Defined Rewrite Rules and reading flow.**
>
> We sincerely thank Reviewer A6nL for the time and effort devoted to carefully reading our manuscript and for providing constructive feedback. We have addressed all of your comments in the revised PDF.
>
> * **1. Dependence on Manually Defined Rewrite Rules and Reading Flow.**
>
>   As Reviewer A6nL pointed out in the comments, automatically extracting important equalities or rewrite rules from data or from the task structure is a promising direction that may further improve performance. We agree this is an exciting direction, and we believe that our current tool could be extended with such automatically discovered rules in future work.
>
>   In this work, our focus is on developing a unified framework that makes modern symbolic regression algorithms aware of symbolic equivalence. Each rewrite rule in our system is constructed from known mathematical equalities; in other words, we assume these equalities are given and study how to encode them uniformly using e-graphs and leverage them during learning.
>
> * **2. Writing Concerns.**
>
>   We have moved the position of Figure 1 and added color-highlighted sentences immediately after Section 3 to improve the reading flow. The annotated result of Table 1 has been updated in the revised PDF.

---

### Official Review · Reviewer_Yckw · 2025-11-01

**Soundness:** 1
**Presentation:** 1
**Contribution:** 2
**Rating:** 2
**Confidence:** 5

**Summary:**

This paper introduces the Equality Graph (e-graph) into the symbolic regression (SR) framework to enable uniform rewards and loss assigned to the mathematical equivalent expressions, enabling a more efficient and robust SR search. The paper provides both theoretical analyses and experiments on several backend learning/search methods to support the claim.

**Strengths:**

1. **Significance**: The paper focuses on an important and fundamental problem in symbolic regression (SR), and the introduced e-graph is reasonable for equivalence-aware SR frameworks.

2. **Multiple Backends:** The paper conducts e-graphs on several learning/search methods (MCTS, DRL, and LLM) to show a universal advantage of capturing expression equivalences in SR methods.

**Weaknesses:**

1. **Novelty:** Though the e-graph is new in SR, similar ideas of using graph-based representations to capture expression equality have already been studied for SR. For example, Expression DAGs [1] include the same idea of sharing sub-expressions in DAGs for expression equality. The paper lacks a sufficient literature review on this topic to differentiate its novelty and contribution.

2. **Potentially Biased UCT Search**: The proposed EGG back-propagation in MCTS might lead to biased UCT selections, see Question 2.

3. **Experimental Soundness:** The whole experimental section can be regarded as an ablation study, which only focuses on implementing typical baselines with or without EGG on datasets with abundant equivalencies. However, benchmarkings on widely adopted general datasets (e.g., SRBench [2]) with relevant prevailing baselines (e.g., SPL [3] for MCTS and uDSR [4] for DRL) are missing, which cannot support the claim "EGG-SR ... discovering equations with lower normalized mean squared error than *state-of-the-art* methods" in the abstract. Especially, the *state-of-the-art* methods could include many transformer-based methods, such as TPSR[5], RAG-SR[6], and also DGSR [7], an E2E transformer model to encode expression equivalences.

4. **Clarity:** There are some confusing definitions and notations, see Question 1.

[1] Kahlmeyer, Paul, et al. "Scaling up unbiased search-based symbolic regression." Proceedings of the Thirty-Third International Joint Conference on Artificial Intelligence. 2024.

[2] La Cava, William, et al. "Contemporary symbolic regression methods and their relative performance." Advances in neural information processing systems 2021.DB1 (2021): 1.

[3] Sun, Fangzheng, et al. "Symbolic Physics Learner: Discovering governing equations via Monte Carlo tree search." The Eleventh International Conference on Learning Representations.

[4] Landajuela, Mikel, et al. "A unified framework for deep symbolic regression." Advances in Neural Information Processing Systems 35 (2022): 33985-33998.

[5] Shojaee, Parshin, et al. "Transformer-based planning for symbolic regression." Advances in Neural Information Processing Systems 36 (2023): 45907-45919.

[6] Zhang, Hengzhe, et al. "RAG-SR: Retrieval-augmented generation for neural symbolic regression." The Thirteenth International Conference on Learning Representations. 2025.

[7] Holt, Samuel, Zhaozhi Qian, and Mihaela van der Schaar. "Deep Generative Symbolic Regression." The Eleventh International Conference on Learning Representations.

**Questions:**

1. **Clarity**
- In line 42, "A promising yet underexplored direction for reducing the *effective* search space", do you mean reducing the ineffective or redundant search space?
- *Inconsistences*: In line 83, coefficients are denoted as $\boldsymbol{c}$, then in line 93 they are denoted as $const$, and in line 213 $c$ is referred to constant to balance exploration and exploitation.
- *Confusions*: In Fig.1 and lines 133, 134, 252, and 253, notations explaining the substitution rule for recursively constructing a closed-form expression all use a single notation $A$ for a place-holder subexpression, making it hard to distinguish their hierarchy structure and correspondence in the figure. Using a subscription to distinguish placeholders might be better for clarity.
- *Missing Critical Definitions*: There are no formal definitions of states and actions in the Markov Decision Process (MDP) for both MCTS and DRL in the main context. There is also no clear definition of what a node $s$ represents in line 211 (I assume it should be the candidate expression $\phi$ inferred from the example in line 252) and what a sequence $\tau$ represents in line 272 (Is it the same as $\phi$ for node $s$? How EGG is represented as a state here?).
2. **EGG-based Backpropagation**: I have two major concerns about the EGG-based backpropagation based on my understanding, both of which arise from the backpropagation to all the equivalent sub-expressions in MCTS:

- The EGG-based backpropagation seems to backpropagate the increasing visit counts to all the equivalent sub-expression generation paths (even if they are not explored at this stage). Does it mean that the visit counts are dependent on the number of equivalents a sub-expression has, so that visit counts of an expression with more equivalents will grow faster, leading to potentially under-explored UCT search?
- Even if the above case is fine for the terminal nodes that are equivalent, the non-terminal nodes on the intermediate paths might not be equivalent. For example, two equivalent subexpression, their paths $sin^2(A_1)+cos^2(A_2) \rightarrow sin^2(x)+cos^2(x)=1$ and $\frac{B_1}{B_2} \rightarrow \frac{y}{y} = 1$  can have distinct intermediate expressions $sin^2(A_1)+cos^2(A_2)$ and $\frac{B_1}{B_2}$ if the placeholders $A_i$ and $B_i$ are not further sampled to be the same variables respectively. In such a case, would the synchronization of the equivalent terminals' backpropagation regarding the visit counts also affect these non-equivalent intermediate nodes (i.e., more number of visits if they have equivalent terminals), so that they might have biased UCT selections?

I'm happy to change my score if the above concerns and questions are properly addressed.

---

> ### Author Response · Authors · 2025-11-20
> **Thanks reviewer Yckw for the feedback and clarifications on EGG-based propagation, grammar definition and imprecise sentences.**
>
> We sincerely thank the reviewer Yckw for the time and effort devoted to carefully reading our manuscript and providing detailed, constructive feedback. We have carefully addressed all of your points in the revised PDF.
>
> **1. Bias in UCT-based Search using EGG-based propagation**
>
> Our update strategy is inspired by prior work that uses a *transposition table* to explicitly store identical nodes in the search tree [1, 2]. These tables record all identical states via hashing-based methods and have been shown to be effective in domains such as Go and Hearthstone. In symbolic regression, however, two nodes in the search tree can only be recognized as identical up to symbolic equivalence induced by rewrite rules, so a standard hashing-based transposition table cannot be directly applied.
>
> The backpropagation procedure in EGG-MCTS differs from that of vanilla MCTS, but it does **not** introduce bias, according to existing research on theoretical analysis of the impact transposition table on MCTS.
>
> Specifically, the interpretation of $\mathtt{visits}(s)$ in Equation 1 changes: it no longer counts how many times this specific tree node appeared on a simulated path, but instead counts how many times any representative of its equivalence class was visited. Conceptually, this is equivalent to using a transposition table in MCTS that shares statistics across identical nodes. Such sharing accelerates learning while preserving an unbiased estimation of value.
>
> [1] Transpositions and move groups in Monte Carlo tree search. CIG, 2008.
>
> [2] Enhancing Monte Carlo Tree Search for Playing Hearthstone. CoG, 2019.
>
> ---
>
> **2. A clarifying example for the suggested rewrite rules for equalities**
>
> We have added another example of  rewrite rules using the suggested two equality rules:   $\sin^2(a)+\cos^2(a)\leadsto 1$ and also $a/a \leadsto 1$,
>
> It is included in the revised paper [https://openreview.net/pdf?id=oh9ChF7Pv0#page=24](https://openreview.net/pdf?id=oh9ChF7Pv0#page=24)
>
> These examples clarify how EGG uses rewrite rules to identify and merge symbolically equivalent expressions within the e-graph.
>
> ---
>
> **3. Grammar definition of symbolic regression**
>
> We have also added a concrete example of how expressions are generated from our grammar:
>
> [https://anonymous.4open.science/r/egg-sr/plots/expr_as_rules-crop.png](https://anonymous.4open.science/r/egg-sr/plots/expr_as_rules-crop.png)
>
> Also see page 2 of the paper (Figure 1) for a detailed illustration of how the grammar definition is applied to generate an expression:
> [https://openreview.net/pdf?id=oh9ChF7Pv0#page=2](https://openreview.net/pdf?id=oh9ChF7Pv0#page=2)
>
> Our grammar representation is more general than a standard expression tree because it can encode the *composition* of sub-expressions, e.g., a rule such as
> ( A \to \sin(x_1) + \cos(A) ).
> This makes it straightforward to extend our framework to recent work such as *Symbolic Regression with a Learned Concept Library* (NeurIPS 2024).
>
> ---
>
> **4. MDP definition for MCTS and DRL**
>
> We have now explicitly included the MDP definitions used in our methods:
>
> * For MCTS, the MDP definition is given on page 24 of the paper:
>   [https://openreview.net/pdf?id=oh9ChF7Pv0#page=24](https://openreview.net/pdf?id=oh9ChF7Pv0#page=24)
>
> * For DRL, the MDP definition is given on page 25 of the paper:
>   [https://openreview.net/pdf?id=oh9ChF7Pv0#page=25](https://openreview.net/pdf?id=oh9ChF7Pv0#page=25)
>
> These clarifications make the underlying decision processes explicit for both components.
>
> ---
>
> **5. Confusion between symbols “c” and “const”**
>
> We agree that the distinction between “c” and “const” was confusing. We have revised the notation throughout the paper to consistently use a single symbol and clarified its meaning, thereby avoiding ambiguity.
>
> ---
>
> **6. Statement about SOTA results**
>
> We acknowledge that the original statement was imprecise. It has been corrected to:
>
> > “EGG-SR consistently enhances a class of modern symbolic regression algorithms across multiple benchmarks, discovering equations with lower normalized mean squared error.”
>
> We have updated the abstract accordingly and highlighted the revised text in color on page 1 of the revised manuscript:
> [https://openreview.net/pdf?id=oh9ChF7Pv0#page=1](https://openreview.net/pdf?id=oh9ChF7Pv0#page=1)
>
> **7. Comparison and analysis with recent SR baselines and SRbench**
>
> We analyze and compare with the listed recent SR baselines in section 3.3 [https://openreview.net/pdf?id=oh9ChF7Pv0#page=7](https://openreview.net/pdf?id=oh9ChF7Pv0#page=7). We have used the Feynman dataset from the SRbench in the appendix  [https://openreview.net/pdf?id=oh9ChF7Pv0#page=31](https://openreview.net/pdf?id=oh9ChF7Pv0#page=31).
>
> Due to each paper building on a different codebase, we will add more comprehensive experimental analysis in the next revision.
>
> Your comments have helped us clarify the presentation, correct imprecise statements, and better explain several technical aspects of our method.

---

> > ### Author Response · Authors · 2025-11-22
> >
> > Dear reviewer, we hope our rebuttal has fully addressed all of your concerns. If you still have remaining concerns, please do let us know. We are very glad to bring more clarifications!

---

### Public Comment · ~Fabricio_Olivetti_de_Franca1 · 2025-11-17
**Important Clarifications on E-graphs and Symbolic Regression in Prior Work**

Dear Authors,

As one of the authors of a closely cited related work [1] and a researcher actively involved in the field of e-graphs and SR, I would like to provide feedback on your paper. I believe a few points concerning the foundational mechanisms of our work could be clarified, especially regarding the scope and novelty of the *eggp* algorithm [1].

A misunderstood point about our work comes from this sentence:
"Prior research (de Franca & Kronberger, 2025) has primarily used rewrite rules to simplify expressions, typically rewriting a
longer expression into a shorter one."

This understates the contribution and primary mechanism of our work. The simplification process in *eggp* is a side-effect of a novel search strategy. We exploit e-graphs for enhanced, non-redundant search space exploration by
- Inserting every visited expression into the e-graph.
- Running equality saturation to compute and store some equivalent forms.
- Exploiting this equivalence information to force the production of novel expressions

For example, when proposing a new expression like $(x + 3y)^2$, the e-graph is queried to check if it or any equivalent form (e.g., $x^2 + 6xy + 9y^2$) has been visited. If found, the proposal is "repaired" using existing equivalent forms to create a novel expression.  This efficient process significantly improves exploration capabilities and is precisely what led *eggp* to achieve SotA results across many benchmarks. This is better explained at a blog post https://symreg.at/blog/2025/equality-saturation-and-symbolic-regression/

Regarding the statement:
"our approach employs e-graphs not only for simplified variants but also to generate a richer set of equivalent variants, enabling easy integration into many symbolic regression algorithms."

I would politely argue that our *eggp* algorithm is already using this strategy, as detailed above, by using eqsat to generate a rich set of equivalent forms for novelty generation.
The "integration into many symbolic regression algorithms" is supported by another of work, presented at the same conference, called rEGGression [2] (https://github.com/folivetti/reggression):

- *rEGGression* is a dedicated Python library that enables the user to create, manipulate, and query e-graphs.
- Like EGG-SR, it facilitates the creation of an e-graph from an expression, running equality saturation, and retrieving all equivalent forms.
- Crucially, *rEGGression* offers functionality beyond the scope of EGG-SR, including:
  * Importing e-graphs and equations generated by other SR algorithms.
   * Advanced querying for top-N expressions with filtering by size, complexity, and number of adjustable parameters.
   * Querying expressions by pattern matching (e.g., returning top expressions following the pattern $"v^x"$).
    * Displaying statistics on the quality of the building blocks.

This demonstrates a clear existing solution for flexible integration and advanced model exploration within the SR community.

Some of these features are explained in this blog post:
https://github.com/folivetti/reggression/blob/main/tutorials/blog/post.md

Specific Technical Corrections:

- Conditional Rewrite Rules: equivalence rules such as $log(a \cdot b) = log(a) + log(b)$ are only valid under specific domain constraints (in this case, $a, b > 0$). It is important to note that conditional rules are already supported in the original *egg* library (not to be confused with eggp) and fully implemented in *rEGGression*.
- The statement: "Most existing e-graph implementations are either in Haskell (Willsey et al., 2021) or provided as Python wrappers (Shanabrook, 2024)" is inaccurate.
  * The widely used *egg* library is written in Rust.
  * Implementations also exist in Haskell (*hegg* and *srtree*) and Julia (*Metatheory.jl*).
   * Furthermore, I would argue that many of these libraries are more flexible and general-purpose than libraries specifically constrained for SR, such as the proposed on.
- Space Complexity: while e-graphs are excellent at compacting equivalence relationships, it is vital to acknowledge that they still suffer from the risk of exponential growth in memory usage. Depending on the starting expression and the set of equivalence rules (particularly common simplification rules like $a + 0 \Rightarrow a$ or $a(b + c) \Rightarrow ab + ac$), the e-graph can quickly "explode." It is possible this behavior was not observed in your experiments due to the omission of some of these "problematic" rules.

I hope these clarifications help to improve the accuracy and depth of your paper.

Best regards

[1] Fabrıcio Olivetti de Franca and Gabriel Kronberger. Improving genetic programming for symbolic
regression with equality graphs. In GECCO. ACM, 2025.

[2] de França, Fabrício Olivetti, and Gabriel Kronberger. "rEGGression: an Interactive and Agnostic Tool for the Exploration of Symbolic Regression Models." Proceedings of the Genetic and Evolutionary Computation Conference. 2025.

---

> ### Author Response · Authors · 2025-11-20
> **Thanks for the clarification of your research on e-graphs and SR and suggested corrections**
>
> Thanks again for clarifying your research. Your contributions to EGG and symbolic regression are very important to the community. I have incorporated your valuable feedback into the revised paper.
>
> I would like to clarify the main differences between your research and our work, as I currently understand them:
>
> * **Our main contribution: symbolic-equivalence-aware learning accelerates the symbolic discovery.** Modern SR algorithms typically treat different output sequences as distinct expressions. In our paper, we use EGG to enforce that the model treats *symbolically equivalent* sequences as identical outputs, and we provide **theoretical justification for why this approach accelerates convergence**.
>
>
> * **Representation: grammar-based vs. expression-tree-based.** Our EGG operates on grammar-based expressions, while your work is based on expression-tree representations. We find the grammar-based formulation more flexible: for example, we can define a composed sub-expression as a single production rule, such as $A \to \sin(x_1)+\cos(A)$. This makes it straightforward to extend our framework to recent work (Symbolic Regression with a Learned Concept Library, NIPS 2024).
>
> * **Implementation and interface.** Your excellent work provides rich functionality for symbolic regression. Yet, the core EGG module is still a Python wrapper around the Haskell implementation ([https://github.com/folivetti/reggression/blob/main/src/Commands.hs](https://github.com/folivetti/reggression/blob/main/src/Commands.hs)). In contrast, our proposed EGG module is designed specifically for grammar-based symbolic expressions and is implemented purely in Python ([https://anonymous.4open.science/r/egg-sr/src/equality_graph/equality_graph](https://anonymous.4open.science/r/egg-sr/src/equality_graph/equality_graph)). The code document is in [https://openreview.net/pdf?id=oh9ChF7Pv0#page=17](https://openreview.net/pdf?id=oh9ChF7Pv0#page=17), which has substantial differences compared to your work. Although our Python EGG library still has substantial room for optimization (for example, by incorporating the conditional rewrite rules you mentioned), it already offers an easier starting point for follow-up research to extend and modify.
>
>
> * **Use of EGG for simplification and equivalence classes.** EGG can also be used to simplify expressions and obtain classes of symbolically equivalent variants. These APIs are similar in spirit to your work. When I started this project, I was only aware of your first paper [1]; the new paper you pointed out is still very recent to me (and I see the repository is being actively updated during the review stage), making it hard for me to summarize your contribution. I have revised our manuscript based on your suggestions to improve its accuracy and have highlighted the modified parts in color.
>
>
> Again, thank you for raising these concerns. If I have misunderstood or missed any important aspect of your work, I would be very happy to address further questions or make additional clarifications.

---

> ### Public Comment · ~Fabricio_Olivetti_de_Franca1 · 2025-11-29
>
> Thank you for going through my suggestions and implementing them in your paper. I still would like to further clarify some points:
>
> > de França & Kronberger (2025) incorporates e-graphs into genetic programming (GP) to
> detect and eliminate redundant individuals, encouraging the GP method to explore novel expression.
>
> like in your proposal, our search algorithm treats equivalent expressions as the same. More than that, the algorithm does not eliminate redundant individuals, it **not even produces them**. We use the e-graph theoretical guarantees **to produce non-redundant** solutions. Our technique is much more powerful than using e-graphs to help the search, it uses an e-graph as the **core** of the algorithm.
> A more accurate description, if I may, would be something like "de França & Kronberger (2025) proposed eggp, the first search-based algorithm for symbolic regression using e-graph as the main core. This algorithm stores the whole history of visited solution in a single e-graph while running equality saturation to generate the equivalent expressions. It then uses the historical information to propose never before visited solutions. In de França & Kronberger (2025) [1], they proposed SymRegg, exploiting the same core concept but dropping the population-based concept. This leads to a search algorithm with the same properties as a random search of being optimal and complete."
>
> > A recent follow-up work rEGGression (de Franc¸a & Kronberger, 2025) provides a richer API
>
> small issue: this is not a follow-up as it was published at the same time :-)
>
> > Our EGG operates on grammar-based expressions, while your work is based on expression-tree representations.
>
> I could argue that in our case, it is easy to translate one into the other. The rEGGression library has a method that shows the "patterns" of an expression, that is basically all the possible transition functions to reach that expression (but granted yours is specifically crafted for this representation)
>
> Finally, I'm currently working on benchmarking the algorithms you are comparing with in this paper (E2E, SPL, LLM-SR) on a more standard benchmark protocol. If you could provide some scikit-learn compatible wrappers to run the EGG-SR methods in any dataset, I can test it in this benchmark. We could have a better understanding of how much your approach improves upon the original algorithms.
>
> Thank you again for your prompt response. Hope this contributes to your research.
>
> [1] de Franca, Fabricio Olivetti, and Gabriel Kronberger. "Equality Graph Assisted Symbolic Regression." arXiv preprint arXiv:2511.01009 (2025).

---

### Author Response · Authors · 2025-11-29
**Summarized review and rebuttal for PC, AC and SAC.**

Dear Program Committee, Senior Area Chairs, and Area Chairs,

We sincerely appreciate the time and effort that the PC, ACs, SACs, and reviewers have devoted to evaluating our submission. The constructive feedback has been invaluable in helping us strengthen the work.

For your convenience, we provide below a summary of the key concerns from the reviewers and our response. The feedback has been incorporated into the revised manuscript.

---
### 1. Problem setting clarification (Reviewer Yckw)

- **Grammar definition for symbolic expressions.**
   We added a concrete example showing how an expression is generated from the grammar, with a step-by-step illustration in the new Figure 1.

- **Clarifying examples of equality rewrite rules.**
   We added illustrative examples of the suggested equality rules in page 24 and showed how EGG uses them to detect and merge symbolically equivalent expressions in the e-graph.

-  **Missing definition of MDP for MCTS and DRL.**
   We now provide explicit MDP definitions for both MCTS and DRL in the appendix (Pages 24–25).

---
### 2. Concern on the proposed algorithm (Reviewer Yckw, public commenter [Fabricio Olivetti de Franca](https://openreview.net/profile?id=~Fabricio_Olivetti_de_Franca1))

- **Bias in UCT-based search with EGG-based propagation.**
   We clarified that our update rule is equivalent to prior work using a transposition table that shares statistics across symbolically identical states. Prior research on MCTS with transpositions shows that this yields an *unbiased* value estimator while accelerating learning.

- **Novelty and comparison with prior e-graph / SR work.**
   We clarified that our main contribution is *symbolic-equivalence–aware learning* that treats all symbolically equivalent sequences as a single target, with theoretical guarantees (a tighter regret bound for MCTS and a lower-variance gradient estimator for DRL).

- **Extra requirements when applying rewrite rules.**
   We clarified that each rewrite rule is defined only on its feasible domain; in SR practice, extreme or out-of-domain cases typically indicate expressions that do not fit the data and often lead to numerical issues such as (-\infty) or NaN. This does not affect our empirical or theoretical results.

---
### 3. Experiment clarification (reviewer b8v3, A6nL, public commenter [Fabricio Olivetti de Franca](https://openreview.net/profile?id=~Fabricio_Olivetti_de_Franca1))

- **Need for a new implementation despite existing e-graph libraries.**
   We clarified that existing systems in Rust, Haskell, Julia, and Python-based wrappers are powerful but not tailored to grammar-based symbolic expressions tightly coupled with sequential decision-making (MCTS/DRL/LLMs). We therefore implemented EGG-SR in pure Python (without Rust/Haskell dependencies), supporting grammar-based production rules and seamless integration with modern learning pipelines.

- **Impact of the choice and size of rewrite rules.**
   We explain that we intentionally avoid rules that cause unbounded growth. Adding more rules expands the e-graph and increases computational cost.

- **Rewrite rule configuration for Table 1.**
   We clarify that the rules for Table 1 are manually constructed for general trigonometric-family expressions, list them in Table 3 of the appendix, and provide the implementation link. We also explain the qualitative effect: missing key rules force more exploration, whereas including them allows EGG to collapse equivalent expressions earlier and speed up learning.

- **Configuration of EGG-LLM.**
   We now specify that EGG-LLM samples (K = 10) sequences from the e-graph, convert them into lambda functions, and pass them as additional candidate expressions (with associated fit scores) to the LLM, in a spirit similar to LLM-SR. This heuristically informs the LLM that many syntactically different expressions can represent the same function.

- **Interpretation of “search tree size” in Figure 3 (Left).**
   We clarify that “search tree size” is measured as the *total number of visits/updates* during learning, not the number of distinct nodes.

- **Dependence on manually defined rewrite rules.**
   We agree that learning or mining rewrite rules is a promising extension and explicitly highlight this as future work. In this paper, we focus on a unified framework that assumes mathematically valid equalities are given and studies how to encode them via e-graphs and exploit symbolic equivalence during learning.

### 4. Related work discussion (Reviewers b8v3 and Yckw)
- **Missing comparison with end-to-end deep SR approaches, SymNet-based methods, and recent SR baselines / SRbench.**
   We expanded Section 3.3 to discuss end-to-end deep SR methods and SymNet, analyze and compare with recent SR approaches, and report results on the Feynman subset from SRbench in the appendix.


Sincerely,

Authors of Submission #10593

---

### Meta-Review · Area_Chair_JRDn · 2026-01-09

**Summary:**

- Novelty and positioning vs. prior SR equivalence work (Yckw, b8v3): While using e-graphs in SR is viewed as promising, reviewers question incremental novelty relative to prior equality-aware representations (e.g., Expression DAGs) and request a stronger, more complete related-work section and clearer differentiation. Several recent SR methods (transformer/generative/DRL, e.g., TPSR, RAG-SR, DGSR, SPL, ...) are missing for comparison/context.

- SOTA claim not sufficiently supported (Yckw, b8v3): Concerns that experiments look like ablations on equivalence-rich datasets rather than broad benchmarking. Missing evaluations on standard suites (e.g., SRBench) and against prevailing baselines.

- Potential algorithmic bias in MCTS due to EGG backpropagation (Yckw): Major technical concern that synchronizing/backpropagating visit counts across equivalent expressions could bias UCT selection, potentially favoring expressions with more equivalents.

- Clarity/notation and missing formal definitions (Yckw): Requests clearer terminology, consistent notation, and formal MDP state/action definitions for MCTS/DRL. Confusing placeholder notation in examples/figures, plus minor presentation issues.

- Dependence on manually curated rewrite rules; robustness and scalability (A6nL, b8v3): Framework effectiveness relies on a hand-designed rewrite rule set. Reviewers ask how rules were selected, how performance degrades when rules are omitted, and whether rule engineering limits generality. They also ask for a scaling analysis of equality-saturation/EGG construction time as a function of expression complexity and number of rules.

- Apparent contradiction in MCTS behavior and pruning claim (b8v3): Figure suggests EGG-MCTS explores more nodes than standard MCTS, seemingly conflicting with the “pruning redundant exploration” claim; needs clarification of what “pruning” means under equivalence merging and how search-tree size should be interpreted.

To address the above concerns, the authors have provided a point-by-point response. After reading this, I think most concerns have been properly addressed. Overall, this paper makes a valuable contribution by enhancing SR through a symbolic-equivalence-aware framework.

**Reviewer Concerns:**

For A6nL and b8v3, I think their concerns have been addressed.

For Yckw, I think most concerns regarding the novelty, claim support, potential bias issue, and presentation issue have been addressed. However, for comparing more SOTA baselines, the authors pointed out the challenge of evaluating under the same setting, which seems acceptable.

**Reviewer Scores:**

For A6nL and b8v3, I think their scores might remain the same.

For Yckw, I think the score might be 4 or 6.

---

### Decision · Program_Chairs · 2026-01-26

Accept (Poster)